



# Magnitude, frequency and climate forcing of global volcanism during the last glacial period as seen in Greenland and Antarctic ice cores (60-9 ka)

Jiamei Lin[1], Anders Svensson[1], Christine S. Hvidberg[1], Johannes Lohmann[1], Steffen
Kristiansen[1], Dorthe Dahl-Jensen[1,5], Jørgen P. Steffensen[1], Sune O. Rasmussen[1], Eliza Cook[1],
Helle Astrid Kjær[1], Bo M. Vinther[1], Hubertus Fischer[2], Thomas Stocker[2], Michael Sigl[2],
Matthias Bigler[2], Mirko Severi[3], Rita Traversi[3], Robert Mulvaney[4]

[1]Physics of Ice, Climate and Earth, Niels Bohr Institute, University of Copenhagen, 2100, Denmark

[2]Climate and Environmental Physics, Physics Institute & Oeschger Center for Climate Change Research,
University of Bern, Sidlerstrasse 5, Bern, Switzerland

[3]Department of Chemistry, University of Florence, Florence, Italy

[4]British Antarctic Survey, Cambridge, UK

[5]Centre for Earth Observation Science, University of Manitoba, Winnipeg, Manitoba, Canada R3T 2N2

Corresponding authors: Jiamei Lin (jm.lin@nbi.ku.dk), Anders Svensson (as@nbi.ku.dk)

**Abstract.** Large volcanic eruptions occurring in the last glacial period can be detected in terms of their deposited sulfuric acid in continuous ice cores. Here we employ continuous sulfate and sulfur records from three Greenland and three Antarctic ice cores to estimate the emission strength, the frequency and the climatic forcing of large volcanic eruptions that occurred during the second half of the last glacial period and the early Holocene, 60-9 ka years before AD 2000 (b2k). The ice cores are synchronized over most of the investigated interval making it possible to distinguish large eruptions with a global sulfate distribution from eruptions detectable in one hemisphere only. Due to limited data resolution and to a large variability in the sulfate background signal, particularly in the Greenland glacial climate, we only detect Greenland sulfate depositions larger than 20 kg km$^{-2}$ and Antarctic sulfate depositions larger than 10 kg km$^{-2}$. With those restrictions, we identify 1113 volcanic eruptions in Greenland and 740 eruptions in Antarctica within the 51ka period - where the sulfate deposition of 85 eruptions is defined at both poles (bipolar eruptions). Based on the relative Greenland and Antarctic sulfate deposition, we estimate the latitudinal band of the bipolar eruptions and assess their approximate climatic forcing based on established methods. The climate forcing of the five largest eruptions is estimated to be higher than -70 W m$^{-2}$. Twenty-seven of the identified bipolar eruptions are larger than any volcanic eruption occurring



in the last 2500 years and 69 eruptions are estimated to have larger sulfur emission strengths than the VEI-7
Tambora eruption that occurred in Indonesia in 1815 AD. The frequency of eruptions larger than the typical
VEI-7 (VEI-8) eruption by the comparison of sulfur emission strength is found to be 5.3 (7) times higher than
estimated from geological evidence. Throughout the investigated period, the frequency of volcanic eruptions is
rather constant and comparable to that of recent times. During the deglacial period (16-9 ka b2k), however, there

is a notable increase in the frequency of volcanic events recorded in Greenland and an obvious increase in the
fraction of very large eruptions. For Antarctica, the deglacial period cannot be distinguished from other periods.
These volcanoes documented in ice cores provide atmospheric sulfate burden and climate forcing for further
research on climate impact and understanding the mechanism of the Earth system.

## 1. Introduction

The dispersal of gas, aerosols and ash particles by volcanic eruptions play a major role in the climate system
(Gao et al., 2007; Robock, 2000). Large volcanic eruptions injecting sulfuric gases into the stratosphere and
forming sulfate aerosols have a global or hemispheric cooling effect of several degrees lasting for several years
after the eruption (Sigl et al., 2015).

Estimations of volcanic stratospheric sulfur injections and of the timing and frequency of large volcanic
eruptions is essential for the ability to understand and model past and future global climate conditions
(Timmreck et al., 2016). For the last 1200 to 2500 years, the ice-core based volcanic forcing records derived
from Greenland and Antarctica (Crowley and Unterman, 2013; Gao et al., 2008; Toohey and Sigl, 2017) provide
an essential record for climate model simulations (Jungclaus et al., 2017) supporting detection and attribution
studies (Schurer et al., 2014) including those applied in the IPCC, but so far the global ice-core based volcanic
record of the last glacial period is poorly documented.

### 1.1 Ice-core records of volcanic sulfate deposition

Several studies have reconstructed the volcanic sulfate deposition for part or all of the Holocene in Greenland
(Cole-Dai et al., 2009; Gao et al., 2008; Sigl et al., 2013) or in Antarctica (Kurbatov et al., 2006; Castellano et
al., 2004; Plummer et al., 2012; Nardin et al., 2020; Cole-Dai et al., 2021). Sigl et al. (2015) applied accurately
dated ice cores synchronized between the two hemispheres to reconstruct global volcanism over the last 2500
years. This so-called bipolar synchronization allows to distinguish large global eruptions from those of
hemispheric or more regional impact. During the last 2500 years, they identified 50 global (bipolar) volcanic



eruptions, 5 of which had a sulfur emission strength large or similar to the Tambora eruption occurring in Indonesia in 1815 CE. Prior to the Holocene no bipolar volcanic sulfate deposition record is currently available from ice cores.

One conclusion drawn from historical eruptions is that there is a significant variability of the same volcanic event in the sulfate deposition as determined in different ice cores both on a regional and a local scale (Sigl et al., 2014; Gao et al., 2007). Part of this regional variability can be explained by the difference in sulfate deposition fluxes at different locations. For example, in Antarctica where geographical distances are large, the sulfate deposition at a specific site will be strongly dependent on the location of the eruption, governing wind patterns, seasonality, etc. Another reason for the lateral sulfate deposition variability, however, is that the deposition is snow-accumulation dependent. Higher snow-accumulation generally leads to higher sulfate deposition and low snow-accumulation may lead to reduced or even absent sulfate deposition caused by post depositional processes on the snow surface, such as wind erosion (Gautier et al., 2016). The spatial variability of sulfate deposition in Antarctica was studied at 19 sites covering the past 2000 years by Sigl et al. (2014), and here both accumulation and post-deposition effects were found to be important factors. In particular, on the East Antarctic plateau where snow-accumulation is very low, the sulfate deposition is lower than at more coastal and higher accumulation sites in Antarctica. The snow accumulation effect is also observed in Greenland (Gao et al., 2007), although the effect is much less pronounced here, because the accumulation rates are more similar in Central Greenland than in different parts of Antarctica. In order to reduce the accumulation bias, Gao et al. (2008) selected five large low-latitude volcanic events from 53 Greenland ice cores and calculated the mean ratio of deposition in individual ice cores; they then applied the deposition ratio between different cores to correct the sulfate deposition for all events in all cores. In general, it is clear that more robust volcanic deposition patterns can be obtained when larger sets of ice cores are included, preferably ice cores from high-accumulation sites should be applied (Gao et al., 2007; Sigl et al., 2014).

One complication related to the derivation of volcanic sulfate deposition in ice cores is the thinning of the ice layers with increasing depth and age. Due to the continual flow of the ice sheets, the annual layers and thus the volcano-derived sulfate deposition becomes thinned with depth; an effect that is most pronounced at high-accumulation sites and close to bedrock. In central Greenland, typical thinning rates of annual layers in the 60-10 ka range are 50-90% depending on age and local flow conditions (Johnsen et al., 2001). To calculate the sulfate deposition of a specific eruption a correction for the thinning at the corresponding depth is needed. Thinning functions are obtained from ice flow modelling, and thus, there is a site-specific dependency on accurate flow modelling associated with the sulfate deposition determination. Knowing the spatial variability in



sulfur deposition observed for recent eruptions, one would expect even larger variability for eruptions occurring
during the last glacial period where snow accumulation was generally lower and the thinning of the annual
layers in the ice cores becomes more important.

## 1.2 Studies of the frequency of volcanic eruptions

The volcanic sulfate record of the Greenland GISP2 ice core has been investigated by Zielinski et al. (1997),
who found that there has been increased volcanic activity during the deglacial period (22-8 ka b2k) as compared
to the average activity of the last glacial cycle. This is interpreted as being related to the tectonic isostatic
response to the melting of the large ice sheets during that period. Based on the geological (non-ice-core) record,
Huybers and Langmuir (2009) found that volcanism increased two to six times during the deglacial period, 12-7
ka b2k, as compared to the average level of eruptions during 40-0 ka b2k interval.

In Antarctica, Castellano et al. (2004) determined the frequency of volcanic eruptions over the last 45 ka based
on the EDC ice core. They found a rather constant level of volcanic activity throughout that period except for
the most recent millennia, where the activity shows an increase. Kurbatov et al. (2006) detected volcanic signals
during the last 12 ka in the Siple Dome A ice core from West Antarctica. They found that the number of
volcanic sulfate signals is decreasing with age, possibly related to the relatively low sampling resolution in the
deeper part of that core. Recently, Cole-Dai et al. (2021) applied the high-accumulation WAIS Divide ice core
to determine a fairly constant Holocene eruption frequency with larger-than-Tambora (1815 AD) events
occurring approximately once per millennium.

## 1.3 Volcanic events identified in ice cores with tephra and sulfate peak synchronization

In historical times, the volcanic origin of an ice-core acidity spike may be pinpointed based on a precise dating
of the ice core alone (Sigl et al., 2015). Further back in time, as the uncertainty of both the ice-core dating and
the dating of the volcanic eruptions increases, the origin of a volcanic ice-core layer can only be determined if it
is associated with a volcanic ash (tephra) deposition in the ice (Gronvold et al., 1995). However, tephra layers
are not always coinciding with sulfate peaks and not all tephra layers match volcanic sulfate signals (Davies et
al., 2010). The ice-core volcanic source identification is important as it helps to constrain the magnitude –
interpreted here as sulfur emission strength rather than the mass of material erupted (Pyle, 2015) and the climate
forcing of the eruption and, furthermore, it allows for a more detailed comparison to modelling studies.



In the last glacial period, many Greenland tephra deposits have been associated with Icelandic eruptions while around a dozen of identified tephra layers originate in North America and Eastern Asia (Abbott and Davies, 2012; Bourne et al., 2015; Davies et al., 2014; Cook et al., 2021 (in preparation)). In Antarctica, tephra layers have been identified and associated with eruptions occurring within Antarctica and in the Southern Hemisphere

(Narcisi et al., 2005; Narcisi et al., 2010; Narcisi et al., 2012). Recently, Mcconnell et al. (2017) identified tephra from the long-lasting and Halogen-rich Antarctic Mount Takahe eruption that occurred around 17.7 ka (Hammer et al., 1997). Tephra of the Oruanui eruption from the Taupo volcano in present-day New Zealand has been identified and dated to 25.4 ka in the West Antarctic Ice Sheet Divide ice core (WDC) (Dunbar et al., 2017).

Volcanic eruptions generally do not deposit tephra in both Greenland and Antarctica, so the bipolar synchronization of sulfur spikes in the ice cores is dependent on an alternative matching technique. Svensson et al. (2020) applied annual layer counting in both Greenland and Antarctic ice cores to match patterns of volcanic eruptions leading to the identification of some 80 bipolar eruptions in the 60-12 ka interval. For the Holocene, a bipolar synchronization of volcanic eruptions was released with the AICC2012 time scale (Veres et al., 2013). It

has recently become possible to test if sulfate has indeed reached the stratosphere, which is a prerequisite for being globally distributed, as the sulfate undergoes characteristic isotope fractionation in the stratosphere (Burke et al., 2019; Gautier et al., 2018; Crick et al., 2021; Baroni et al., 2008), but these analyses are still scarce for the Glacial.

### 1.4 Extending the ice-core volcanic record into the last glacial period


Here we extend the ice-core record of sulfate deposition in Greenland and Antarctica by employing sulfate sulfate (or sulfur in one case; sulfate is mentioned for brevity) records from three Greenland and three Antarctic ice cores in the interval 60-9 ka. We investigate the sulfur emission strengths (i.e. defining the climate impact potential) and the frequency of volcanic eruptions detected in either Greenland or Antarctica. For eruptions

identified in both hemispheres, we estimate the climate forcing against modern analogues (Pinatubo 1991AD) and determine the occurrence of very large eruptions.





## 2. **Methods**

### 2.1 Ice-core records

For Greenland we used the North Greenland Eemian Ice Drilling ice core (NEEM) (Dahl-Jensen et al., 2013), the North Greenland Ice Core Project ice core (NGRIP2) (Andersen et al., 2004), and the Greenland Ice Sheet Project 2 ice core (GISP2) (Grootes et al., 1993), and in Antarctica the WDC ice core (WAIS Divide Project Members, 2013; 2015), the EPICA Dome C ice core (EDC) (EPICA community, 2004), and the EPICA Dronning Maud Land ice core (EDML) (EPICA community, 2006).

The sulfate records were obtained by different methods and with different temporal resolution as detailed in the Table S1. For NGRIP the $SO_4^{2-}$ concentration was obtained by a Continuous Flow Analysis system (CFA) (Bigler, Thesis, 2004). For WDC, sulfur was measured by ICP-MS coupled to a CFA system (Sigl et al., 2016). For EDC and EDML, $SO_4^{2-}$ was measured by Fast Ion Chromatography (FIC) coupled to a CFA system (Severi et al., 2015). For GISP2 and for a second NGRIP profile, $SO_4^{2-}$ was obtained from discrete samples by Ion

Chromatography (IC) (Clausen et al., 1997; Mayewski et al., 1993; Siggaard-Andersen, PhD thesis, 2004). The temporal resolution of the sulfate records decreases with ice-core depth and ranges from sub-annual for the CFA profiles to decadal for the lower-resolution discrete profiles (Table S1 and Fig. S1). The high-resolution NGRIP and WDC records have been resampled to annual resolution, that we see as an upper limit for the effective resolution during the oldest part of the analysis in last glacial. As discussed in section 3.4, we are also

resampling these records to lower resolution in order to obtain comparable resolution throughout the investigated period and in order to investigate eruption occurrences among stadial and interstadial periods. Minor data gaps were interpolated using linear interpolation and larger gaps are indicated as missing data. Sulfur and sulfate records were corrected for the sea-salt-sulfur contribution based on sodium concentrations as sea salt tracer by assuming 0.084 for the ratio of Na/S (Bowen et al., 1979), except for EDC. In all cases, the sea-salt

correction is less than 15% of the measured sulfate background signal, giving a slightly change to the non-sea-salt-sulfur. Methanesulfonic acid (MSA) was subtracted from the Antarctic non-sea-salt-sulfate signal in a minor correction (Cole-Dai et al., 2021). In Greenland, the average concentration of MSA over the Holocene and last glacial period is low (Legrand et al., 1997). As there are no continuous MSA records available for the investigated period, we do not make corrections by the MSA records to obtain the non-sea-salt sulfate.

For the 12-9 ka interval, the sulfate deposition is based on the NGRIP and GISP2 sulfate records of Greenland, and the WDC, EDML and EDC sulfate records of Antarctica.





Besides the sulfate records, the depth assignment of the volcanic peaks was occasionally assisted by application of other high-resolution records that also are indicators of volcanic sulfate deposition. Those include the Electrical Conductivity Measurement (ECM) profile, the Di-electric Profiling (DEP) and the liquid conductivity record, that are also available for most of the applied cores (Table S1 for a complete list of records and references).

**2.2 Background signal determination and volcanic peak detection**

To determine the biogenic sulfate background level of the ice-core sulfate records and detect volcanic peaks above this background, we apply methods similar to those applied for Holocene records (Fischer et al., 1998; Karlof et al., 2005; Gao et al., 2007; Sigl et al., 2013). A running median filter with a width of 50-180 years was applied to estimate the non-volcanic background signal. Due to the different depth resolution of the individual records, and because of the highly variable and abruptly changing background levels of the Greenland sulfate levels across DO-events, it has been necessary to apply different filter widths for different records (Fig. S10 (a-y) and Table S1). A second iteration of a reduced running median (RRM) filter using the same filter widths was applied after removal of the spikes identified in the first iteration. For the coarse-resolution NGRIP IC sulfate⁻ record, a continuous background determination was not possible and the background has been estimated manually for selected events only.

For all records, a volcanic detection threshold was estimated as RRM plus three times the running median of absolute deviations from the RRM (RMAD) using the same window widths as for the background determination (Fig. S10 (a-y)). In Greenland, the RMAD value varies strongly across DO-events due to the much greater background variability during stadial periods as compared to interstadials (Table S2). This implies that the volcanic detection may be compromised for the time windows covering the fast transitions from stadials and interstadials (lasting on the order of decades to up to about 150 years, Capron et al. (2021)) when the background variability changes in a short time. Thus, the volcano frequency found in these short intervals may be impacted, but the volcanic frequency outside of these short intervals is not affected. The duration of a volcanic event is estimated from the fraction of the sulfate spike that is above the RRM and the sulfate deposition associated with the event is calculated by integrating the sulfate concentrations above the RRM across the duration of the event. The volcanic sulfate peak area S is calculated as follows:

$$S = \int_{D1}^{D2}(y - RRM)dD \ (D1 < D < D2) * 0.917 \tag{1}$$





where y is the nss-sulfate, 0.917 is the ice-water density ratio, and the sulfate layer in the ice core is constrained
        between the depths D1 and D2. At shallow depth, the temporal duration of the sulfate deposition can be
        determined precisely, but in the last glacial period peak broadening by diffusion is hampering such a
        determination.

**2.3 Correction for ice flow / layer thinning**

As the volcanic layer is buried in the ice sheet, the layer is being thinned by ice flow. In order to calculate the
        amount of sulfate deposited on the ice sheet at the time of the eruption a correction for the thinning of the
        volcanic layer in the ice must be applied.

$$SF = \frac{S}{T}$$                                                                                             (2)

        Where SF is the accumulated sulfate flux and T is the layer thinning. The thinning or the ratio between the
original deposition thickness and the layer thickness in the ice core has been calculated by site-specific ice-flow
        models that we applied here (Table S1 and Fig. S1). During the last glacial period, the thinning of the ice can be
        significant. For the Greenland cores, the thinning ranges from a 60% reduction of the layer thickness at 12 ka
        b2k to as much as 90% at 60 ka b2k (Table S1). For Antarctica, the thinning is most significant for the high-
        accumulation WDC core, where it approaches 95% at 60 ka b2k (Fudge et al., 2016; Buizert et al., 2015),
whereas the EDC core exhibits less thinning of 30% at 60 ka b2k (Fig. S1). For the WDC core, we apply the
        thinning function of Fudge et al. (2016) for the upper 2800 m, whereas below 2800m we apply a simple linear
        fit to the gas-based thinning function (Buizert et al., 2015) that has large and possibly unrealistic wiggles (Fig.
        S1). As the volcanic sulfate deposition scales with the amount of thinning, an inaccurate thinning factor has
        large implications for the calculated sulfate depositions.

It should be noted that the volcanic sulfate flux calculation assumes that the sulfate deposition is mainly
        occurring as dry deposition in the last glacial period. For high accumulation sites, such as those in Greenland
        and coastal Antarctica, it is quite likely that wet deposition is dominating at present day (Kreutz et al., 2000;
        Schupbach et al., 2018). We are not attempting to resolve this issue here, but we are aware that it may have a
        bias in our estimation of volcanic sulfate flux.



## 2.4 Correction of volcanic signals for low resolution data

As the depth resolution of the sulfate records for the GISP2 and NEEM cores is relatively low (Fig. S1), adjacent acidity peaks may be merged into falsely large acidity spikes. We made a manual correction for this effect by comparing to the corresponding higher-resolution ECM and DEP records of the same core and removed or split falsely large peaks according to the associated ECM or DEP peaks.

## 2.5 Volcanic sulfate deposition records

We generate three lists of volcanic sulfate deposition for the 60-9 ka period: one for eruptions identified in the Greenland ice cores (Table S3), one for eruptions identified in the Antarctic ice cores (Table S4), and one for global (bipolar) eruptions identified in both Greenland and Antarctica (Table S5). The bipolar list contains the large global eruptions identified by Svensson et al. (2020) and includes two large bipolar eruptions of Veres et al. (2013) for the 12-9 ka period. Furthermore, two additional bipolar eruptions at 44.75 ka b2k and 44.76 ka b2k have been included, applying the same methods as described in Svensson et al. (2020). For the bipolar list, there is no bipolar eruptions have been attributed in the 24-16 ka period, where ice core synchronization is difficult, so the list covers an effective period of 42 ka. The list of Northern Hemisphere (NH) volcanism contains all volcanic deposits larger than 20 kg km$^{-2}$ in any of the applied Greenland ice cores, and the list containing the Southern Hemisphere includes all deposition events larger than 10 kg km$^{-2}$ for the Antarctic ice cores. This cut-off is necessary because of the highly variable sulfate background in the Greenland cold periods of the last glacial predominatly associated with mineral dust (e.g. gypsum (Svensson et al., 2000)) (Table S2). A sulfate deposition of 20 kg km$^{-2}$ corresponds to half the Greenland deposition from the 1815 AD Tambora eruption, thus refers to quite large events in terms of total sulfur injections into the atmosphere. If the cut-off is set at a lower value, a large number of presumably non-volcanic spikes will be identified in stadial periods. A similar cut-off applied to the sulfate deposition record of the last 2000 years (Sigl et al., 2013) reduces the number of identified eruptions in Greenland from 138 to 49 events. Likewise, for Antarctica, we are only identifying events that are referred to as 'large' or 'very large' by Cole-Dai et al. (2021).

When eruptions are detected in several ice cores in Greenland or Antarctica, we provide the sulfate deposition value for each core, the range spanned by all the cores, and the calculated average sulfate deposition value. The average Greenland sulfate deposition is calculated by the simple mean of the three cores due to their similar sulfate-deposition levels and proximity of the ice coring sites (Fig. S2). In Antarctica, however, the ice-core sites are far apart and the EDC core generally has the lowest sulfate deposition due to its low accumulation. Therefore, when there is only a sulfate signal present in one or two cores, the average sulfate deposition of




Antarctica is calculated with a rescaling factor, similar to the method applied by Gao et al. (2008). The scaling factor is based on the relative ratio of sulfate deposition from the 30 largest eruptions (in terms of the sulfate deposition) in the three cores in the period of 60-9 ka. For the 30 largest eruptions identified in Antarctica, the ratio of the sulfate deposition among the three Antarctic cores is EDC:EDML:WDC =0.72:0.87:1.4 (Fig. S2) .

The error estimation of volcanic sulfate deposition is according to the propagation of formula (1) and (2).

Considering the sulfate diffusion and layer thinning of ice cores, we derived the area of sulfate peak and then estimated the uncertainty of the area of sulfate peak and layer thinning. We apply an estimated 21% uncertainty of the sulfate concentration levels independent of the analytical technique, an estimated 10% uncertainty of the applied thinning functions, and a 15% uncertainty related to the varying temporal resolution of the sulfate records. This leads to an error estimate of up to 26% for individual deposition events that we apply as error

estimate for all the derived sulfate depositions.

### 2.6 Latitudinal band assignment of bipolar volcanic eruptions

The volcanic sulfate deposition in Greenland and Antarctica shows a distribution pattern (Table S5 and Table S6), similar to that derived from the aerosol-climate modeling of volcanoes over past 2500 years (Martshall et al., 2020). To estimate the latitudinal band of bipolar volcanic eruptions of unknown origin, we applied the

Support Vector Machine (SVM) classification model of Gapper et al. (2019) and Vapnik (1998). The classification model is based on a kernel function generation and logistic regression (Fig. S3). For an optimal classification, a maximum-margin hyperplane was used to separate two classes. The kernel scale and box constraints were chosen for the model and a Bayesian optimization was used to optimize the above two parameters to yield the best classification model (Fig. S3). The model was trained using 21 eruptions for which

the eruption site is known from tephra deposits in the ice (Table S6), and the bipolar eruptions of unknown origin were predicted into two latitudinal bands – above 40°N (NHHL) and below 40°N (LL or SH) (Table S5). Due to the low number of known volcanoes erupted in the high latitudes of Southern Hemisphere, the method does not allow identification of eruptions potentially located in this region.

### 3 Results

### 3.1 The Greenland volcanic sulfate deposition record (60-9 ka)

The sulfate deposition records derived from the NGRIP, NEEM and GISP2 ice cores are displayed in Figure 1 and Figures S9 (a-y). The background level of the sulfate signal in the Greenland cores is in the range of 50-200





ppb with both the absolute level and the signal variability being much higher in stadial periods than in interstadial periods (Table S2). At the abrupt climate transitions during the last glacial period the sulfate

background changes abruptly, sometimes making it difficult to determine the background level and volcanic detection limits (see above). The higher background level and variability during the stadial periods is partly due to the lower snow accumulation during these periods.

Using a 20 kg km$^{-2}$ deposition threshold 1113 volcanic events are identified in Greenland in total (Table S3). Table S8 shows the number of volcanoes detected from one, two or three Greenland ice cores, respectively.

NGRIP has the highest event detection rate because of its superior depth resolution that becomes increasingly critical in the cold climate of the investigated period where annual layers get down to a few centimeters of layer thickness (Fig. S1d).

The difference in sulfate deposition among the cores reflects to some degree a spatial variability in sulfate deposition within Greenland, but is also strongly influenced by the sulfate record type (IC, FIC, CFA), the data

resolution, the thinning correction made for each site, and possibly the accumulation rates. A comparison of the sulfate deposition between NGRIP, NEEM and GISP2 for the 30 largest volcanoes with identified signals in Antarctica ice cores shows that there is a very large spread in the signal from event to event (Fig. S2 (a-d)). Due to the large variability of sulfate deposition values among the cores, we provide the full range of sulfate deposition values spanned by the cores as well as the average sulfate deposition value, when an eruption is

identified in several ice cores.

Fig. S2 (e) shows a comparison between the 57 largest volcanic events identified in the NGRIP ice core, as derived from the high-resolution CFA $SO_4^{2-}$ record and from the lower resolution IC $SO_4^{2-}$ record, respectively (Table S3). This comparison illustrates the differences in calculated sulfate depositions due to measurement technique and record resolution only, as the sulfate deposition and the thinning factor uncertainties are

eliminated. It is obvious from the comparison that the derived sulfate deposition of a single event in a single core should not be taken at face value; there are very large uncertainties on top of the variability caused by the spatial distribution pattern. Still, being the first systematic compilation of large volcanic eruptions from the last glacial period, the estimated depositions give a good approximation about the order of sulfur emission strength and the frequency of large volcanic eruptions in Greenland.

Out of the 1113 volcanic events identified in Greenland we identify 10 very large events with sulfate deposition estimated to be larger than 300 kg km$^{-2}$ – where 9 have an Antarctic counterpart indicating their global impact. We identify 87 volcanic events with sulfate deposition larger than 135 kg km$^{-2}$, thereby exceeding the Icelandic Laki eruption occurring in 1783 CE, which produced the largest Greenland sulfate deposition in the last 2500





years (Sigl et al., 2015). The largest sulfate deposition in Greenland is by far the North Atlantic Ash Zone
(NAAZ II) event occurring at 55.38 ka b2k (Rutledal et al., 2020; Austin et al., 2004). The sulfate deposition of
this event is in the range of 866-1436 kg km$^{-2}$, which is more than a factor of two higher than any other event
occurring in the investigated period. The second largest event is an unknown eruption occurring at 45.55 ka b2k
with a sulfate deposition in the range 200-849 kg km$^{-2}$. Those and other large events will be discussed in more
detail in section 4.4.

**3.2 The Antarctic volcanic sulfate deposition record (60-9 ka)**

The volcanic sulfate deposition extracted from the EDML, EDC and WDC ice cores are shown in Fig. 1 and
Fig. S10 (a-y). The sulfate background level in Antarctica is comparable to that of Greenland (Table S2), but
there is much less signal variability and an absence of abrupt shifts associated with Greenland DO transitions.
In total, 740 (350) volcanic sulfate deposition values estimated to be larger than 10 (20) kg km$^{-2}$ are identified in
the Antarctic ice cores (Table S4 and Table S8). WDC has the highest accumulation of the three records, but as
the deepest part of the applied WDC record is close to bedrock, layer thinning becomes very significant here
(Fudge et al., 2016) (Table S1). Therefore, in the deepest section of the WDC core, fewer eruptions are detected
as compared to EDML and EDC, and the derived WDC sulfate deposition has a very strong dependency on the
applied thinning function. EDC has the lowest accumulation of the three and also the lowest temporal resolution
of the sulfate record. Therefore, smaller events are generally not detected in the EDC core, and the low
accumulation most likely causes some eruptions to be partly or entirely absent from the record (Gautier et al.,
2016).

The volcanic deposition of the 30 largest events in Antarctica are compared in Fig. S2. In general, there are
large differences among the sulfate deposition in the different cores for the same events, primarily owing to the
large gradient of accumulation rates over Antarctica (Sigl et al., 2014). The influence of different measurement
types (FIC or CFA-ICP-MS) and the temporal resolution of the records may also accounts for some of the
variability, as well as post-depositional processes (Gautier et al., 2018).

In the 60-9 ka period, there are 32 large volcanic events with a sulfate deposition larger than 78.2 kg km$^{-2}$,
which is the largest volcanic deposition in Antarctica over the most recent 2500 years (Sigl et al., 2015). Fourty-
nine volcanic events have a sulfate deposition larger than 63.6 kg km$^{-2}$ corresponding to the Antarctic sulfate
deposition of the Kuwae 1458 CE eruption (Sigl et al., 2015). Among the 50 largest eruptions, 28 are also
identified in Greenland ice cores. The largest volcanic signal detected in Antarctica is occurring at around 46.69
ka b2k with a sulfate deposition of 121.0-369.4 kg km$^{-2}$.



### 3.3 The bipolar volcanic sulfate deposition record (60-9 ka)

For the 60-9 ka period, the 85 bipolar volcanic eruptions that have been identified in both Greenland and Antarctic ice cores (Veres et al., 2013; Svensson et al., 2020) are listed separately in Table S5. We note that no bipolar eruptions have been identified in the interval 24.5-16.5 ka, because of the general difficulty to synchronize ice cores in this period (Seierstad et al., 2014). For the bipolar eruptions the volcanic forcing can be estimated using methods previously applied to Holocene eruptions (see section 4.3). Twenty-eight out of the 85

bipolar eruptions have been identified in all six ice cores. Two of the bipolar volcanic eruptions are known from Greenland ice-core tephra deposits to be of Icelandic origin: The Vedde ash eruption (around 12.17 ka b2k) (Mortensen et al., 2005) and the NAAZ II eruption (around 55.38 ka b2k) (Rutledal et al., 2020). Furthermore, there are tephra deposits in Greenland from the Japanese Towada-H eruption (ca 15.68 ka b2k) (Bourne et al., 2016). In Antarctica there is tephra deposited from the New Zealand Oruanui, Taupo, eruption (ca 25.46 ka)

(Dunbar et al., 2017) (Table S6).

### 3.4. Relationship between data resolution and the volcanic eruption record of the last glacial (60-9 ka)

To investigate the frequency of different size categories of volcanic eruptions over time, we need to consider the sample resolution of the underlying sulfate records (Fig. S1(a+b)). As we go back in time, the layer thinning

becomes stronger which makes it increasingly difficult to detect smaller eruptions signals. Depending on the ice-core sample resolution, the apparent number of smaller eruptions decreases relatively faster with increasing age (or depth) than the larger eruptions as the former will no longer exceed the detection threshold determined by the background signal due to smoothing of the records. The effect is seen when the frequency of detected eruptions is separated into fractions of sulfate deposition sizes. We separated the eruptions according to the 0.7

and 0.9 quantiles of the volcanic sulfate deposition distribution, which for Greenland are 68 kg km$^{-2}$ and 140 kg km$^{-2}$ and for Antarctica are 25 kg km$^{-2}$ and 50 kg km$^{-2}$ (Fig. S4). When considering the frequency of volcanic eruptions across the investigated period, we should preferably have the same or at least a comparable temporal sulfate sample resolution throughout the period in order to minimize this age (or depth) bias.

For Greenland, only the NGRIP sulfate record has high (sub-annual) resolution for most of the investigated

interval (Fig. S1b). Therefore, the number of eruptions detected in different periods will depend strongly on the NGRIP sample resolution. The NGRIP sulfate record is measured in 1mm resolution, but the effective resolution is lower than that due to smoothing of the record in the ice and – most importantly – by signal dispersion during the measurement. Using an approach similar to that of Rasmussen et al. (2005), we performed a spectral analysis of the NGRIP sulfate record to determine a signal cut-off at around 2 cm, which is



comparable to that of other components analyzed with the same setup (Bigler et al., 2011) (Fig. S5). Two cm
corresponds to about 2 annual layers in NGRIP in the climatically coldest and oldest sections of the investigated
interval. In Table S7 and Fig. S4 we investigate the effect of increased smoothing of the NGRIP sulfate record
for the detection of sulfate spikes. As expected the number of detected eruptions decreases with increased
smoothing, however, with 3 years smoothing, the number of larger sulfate spikes starts to increase as adjacent
sulfate peaks are being merged.

As a compromise between compensating for the decreasing resolution of the record with depth and at the same
time avoiding a strong influence of the peak merging effect, we smoothed the NGRIP sulfate record to 2-year
resolution for the eruption frequency analysis, knowing that this is eliminating a fraction of the smaller eruptions
in the younger section of the record. With the 2-year smoothing there is probably still a bias towards a higher
detection rate in the stadial periods due to the high signal variability of the sulfate record in those periods (see
discussion in section 4.1). The GISP2 sulfate record has lower than 10-year resolution in the 60-14 ka period
(Fig. S1), so in this period we can only use the GISP2 sulfate record to estimate the sulfate deposition of the
larger eruptions. The NEEM sulfate record has a constant resolution of 10 years and we apply this record
throughout the investigated period. In Fig. 4 we show the records of detected Greenland eruptions per
millennium with the sulfate deposition grouped into three size fractions.

For Antarctica, EDML and EDC have fairly constant thinning rates for the 60-9 ka period (Fig. S1c), whereas
WDC has very strong thinning in the oldest part of the period, due to the high accumulation rate (Fig. S1c).
Spectral analysis of the Antarctic sulfate records shows the effective resolution (Table S1) and the signal cut-off
is around 2 cm, 3 cm, and 1 cm for EDML, EDC and WDC, respectively (Fig. S5). In order to have comparable
resolution throughout the investigated period, we smoothed the entire WDC sulfate record to 2-year resolution
to homogenize the record resolution across the investigated period. For EDML the correspondingly effective
signal resolution are 1-2 year for the entire period, so we keep the original resolution of the EDML sulfate
record for the entire period. The EDC sulfate record has almost constant resolution, so we apply this record
throughout the investigated period. The result of the frequency analysis is discussed in section 4.1.

**4. Discussion**

In the following we investigated possible climate-volcano links on different time scales, estimated the last
glacial volcanic forcing, and compared to the glacial volcanic record to that of the last 2500 years. Many studies
have discussed a possible impact of climate on the frequency and magnitude of volcanic eruptions (Cooper et



al., 2018). In particular, the melting of the large ice sheets at the end of the last glacial period and the
corresponding sea-level rise are thought to have created crustal stress imbalances and increased the volcanic
eruption frequency in the deglacial period (Watt et al., 2013; Huybers and Langmuir, 2009; Zielinski et al.,
1996; Zielinski et al., 1997).

### 4.1 Millennium scale volcanic eruption variability

We investigate the variability in eruption frequency and sulfur emission strength with the DO cycles by
separating the detected volcanic eruptions according to climate of 'cold' and 'milder' periods, which the onset
and termination of DO events defined by Rasmussen et al. (2014). 'Cold' periods constitute the stadial periods
from 21 ka b2k to 60 ka b2k, covering a total of 21.8 ka period. 'Mild' periods are the remaining periods that
include all of the interstadial periods from 21 ka b2k to 60 ka b2k, adding up to 17.2 ka years. We grouped the
detected eruptions according to the size of deposited sulfate and also investigated the effect of smoothing the
NGRIP sulfate record to 1, 2, 3, and 5 years resolution (Fig. S6).

Fig. 2 summarizes the number of volcanic eruptions per millennium detected by the Greenlandic and the
Antarctic sulfate records in the Greenland cold and mild periods. For Antarctica, the number of detected
eruptions (per millennium) is very similar for NH cold and mild periods, independent of deposition size
categories and record smoothing. For Greenland, however, the number of volcanic eruptions is higher in cold
periods, and this in smaller deposition category (20-68 kg km$^{-2}$) decreases with increasing NGRIP sulfate record
smoothing in the number of detected eruptions in cold and milder periods (Fig. S6).

This observed difference leaves us two options: either there is a true millennium-scale volcanic eruption
variability related to northern hemispheric climate variability, or the dependency is an artefact caused by the
very different properties of the Greenland sulfate record in cold and milder periods. In Greenland, the variability
of the sulfate background level is high in cold periods, stadials and the LGM, and low in milder periods such as
interstadials (see section 3.1). We try to take this difference into account by employing the variability-dependent
RMAD to determine the eruption detection limit, but because of the very different character of the sulfate
profiles in cold and milder periods there is still the possibility of having a climate related detection bias of
sulfate spikes that is unrelated to the actual volcanic eruption record. For example, in cold periods, it could be
that at strong seasonal sulfate spikes associated with marine or terrestrial (non-volcanic) sulfate production
could be detected as smaller volcanic eruptions. For recent times, Gao et al. (2007) has demonstrated that the
volcanic sulfate deposition to some degree is accumulation dependent, with increased accumulation leading to
higher sulfate deposition. Since, however, the highest accumulation occurs in the interstadial periods, this effect



would lead to relatively higher sulfate deposition in interstadial periods, which is the opposite of what is

observed.

We investigated the effect of a possible seasonal sulfate contributions of non-volcanic origin by a smoothing exercise where successively increased smoothing of the NGRIP sulfate record increasingly eliminates the detection of smaller short-lived (seasonal) sulfate spikes. Fig. S4, Fig. S6 and Table S7 show the number of sulfate spikes detected in cold and milder periods as a function of the degree of smoothing. It is seen that for

cold periods, the number of detected small and short-lived sulfate spikes decreases significantly with increased smoothing, whereas the effect is almost absent for milder periods. This suggests that the higher number of detected sulfate spikes in cold periods is mainly due to spikes of periodic duration of sulfate background which we believe are less likely to be of volcanic origin. For volcanic eruptions that are large enough to inject sulfate into the stratosphere modern observations show that the sulfate fallout typically will last several years. We thus

conclude that the apparent increased volcanism detected in cold periods is most likely an artefact caused by non-volcanic seasonal contributions from high-sulfate marine or terrestrial sources. It could for example be contributions from gypsum that is known to have high deposition fluxes in Greenland during stadial periods (Legrand and Mayewski, 1997). Indeed, we do observe a higher correlation in stadial periods than interstadial periods between the volcanic sulfate deposition and both insoluble dust concentration and sodium concentration

for bipolar events (Fig. S9). We cannot exclude, however, that other mechanisms such as different atmospheric circulation patterns, or different sulfate deposition process in terms of dry versus wet deposition, are also affecting the Greenland volcanic detection rates in stadial versus interstadial periods. We note that because an increased smoothing of the sulfate record mostly decreases the number of smaller detected sulfate depositions, the effect of smoothing on the large accumulated sulfate deposition over time is less important (Fig. S4).

**4.2 Long-term volcanic eruption variability**

In order to estimate the long-term pattern of volcanic activity, the accumulated number of volcanic eruptions and resulting sulfate deposition in Greenland and Antarctica over the investigated period are compared in Fig. 3 (same data as applied in Figure 1). The cumulative volcanic number (sulfate deposition) in Greenland is larger than in Antarctica by a factor of 3.2 (4.7), considering only sulfate depositions larger than 20 kg km$^{-2}$ at both

poles. We attributed this mainly to the larger number of volcanoes situated in the NH, but it is possibly also related to the proximity of Iceland to Greenland and other volcanic regions upwind of Greenland, and to the differing atmospheric circulation patterns between the two hemispheres (Toohey et al., 2019). In terms of cumulated volcanic number there is some variability that could be climate related, but overall the eruption





frequency is more constant as compared to the geological record (Brown et al., 2014), most likely because the
ice-core preservation and identification are much more homogeneous over time than that of radiometrically
dated eruptions.

The number of eruptions per millennium shows a fairly constant level both for Greenland and for Antarctica
(Fig. 4, Fig. S7 and Fig.S8). The Early Holocene increase in global volcanism as suggested by Watt et al. (2013)
is not identified in our record. Neither is the periodicity of Campanian eruptions as identified by Kutterolf et al.
(2019) but this pattern could be masked by eruptions from elsewhere.

Based on the EDC sulfate record, Castellano et al. (2004) determined the Antarctic volcanic sulfate deposition
to be fairly constant over the last 45 ka, except for an increase during the last 2000 years. The eruption
frequency level of 5-10 eruptions per millennium determined in that study is on the low side compared to the
present study, in particular for the older part of the period (Fig. 4). This is likely because the EDC sulfate record
has low resolution and could not pick up all of the smaller short-lived eruptions, also because the low
accumulation at the EDC site does not record some eruptions (see methods/results section). Over the 60-9 ka
interval we have detected 224 volcanic eruptions in EDC as compared to 513 and 496 in the higher resolution
WDC and EDML records, respectively (Table S4).

Recently, Cole-Dai et al. (2021) compiled a Holocene volcanic record from the high-resolution sulfate record of
the high-accumulation WDC ice core. The authors determine a fairly constant Holocene volcanic activity with
identification of 426 eruptions of which 162 and 44 have sulfate depositions larger than 10 kg km$^{-2}$ and 30 kg
km$^{-2}$, respectively. Those numbers are very comparable to our findings for Antarctica in the period of 11-9 ka
(Fig 4).

Based on the same GISP2 sulfate dataset as applied in the present study and after introducing a rescaling to
compensate for the decreasing sample resolution with depth, Zielinski et al. (1996, 1997) identified an enhanced
northern hemispheric volcanic eruption frequency in the deglacial period (17-6 ka) compared to the 60-0 ka
average. In our 3-core Greenland sulfate deposition compilation, we also identified an increased volcanic
eruption frequency over the deglacial period, which is 34% higher than the Holocene period (2-0) for the
average volcanic number per millennium (Fig. 2 and Fig. 4). In particular, there is an increased fraction of large
sulfate deposition events in the deglacial period. In Antarctica, neither the detected eruption frequency nor the
total sulfate deposition over the deglacial period are higher than the average for the full 60-9 ka interval. We
therefore conclude that the increased number of large eruptions occurring in the deglacial period are mostly of
northern hemispheric origin. This hypothesis is supported by the elevated number of tephra deposits identified
in Iceland records (Maclennan et al., 2002) and in Greenland during the deglacial period mostly of Icelandic





origin, a few from Japanese and unknow origins (Bourne et al., 2016; Zielinski et al., 1997; Mortensen et al., 2005; Gronvold et al., 1995). Van Vliet-Lanoe et al. (2020) found that the enhanced volcanic events are most likely related to stress unlocking during deglaciation events, which the occurrence of volcanism in Iceland is related to rifting events at the melting margin of the ice sheet. In the period of 14.6-13.1 ka in Southeast Alaska, the enhanced volcanic activity from the Mount Edgecumbe Volcanic Field has also been observed by Praetorius

et al. (2016). The frequency of detected eruptions for the glacial period (60-21 ka) is similar to that of the last 2000 years, when comparing the same sulfate deposition fractions (Fig. 4). We note that for the last 2000 years only eruptions with sulfate depositions larger than 20 kg km$^{-2}$ are included and the sulfate records have been smoothed with a two-year filter. The apparent increase in global volcanism over the last millennia in the geologic records (Brown et al., 2014) is therefore most likely due to an under-representation of older eruption

identifications in that study (Papale, 2018; Deligne et al., 2010). When it comes to sulfur emission strengths of eruptions and accumulated sulfate deposition, however, the last 2000 years appear under-represented in very large eruptions as compared to long periods of the last glacial period (Table 2).

**4.3 Estimating the volcanic forcing of the last glacial 60-9 ka**

To estimate the volcanic forcing from eruptions occurring in the last glacial and early Holocene we applied the

approach of Gao et al. (2007), Crowley and Unterman. (2013) and Hansen et al. (2005). This method was calibrated and evaluated using modern volcanic eruptions (Gao et al., 2008; Crowley and Unterman, 2013), and it is quite likely that the results of the glacial eruptions will be biased by the very different glacial conditions. Calibrated against Pinatubo 1991 (at 15°N), the method is best suited for bipolar eruptions, and it is necessary to separate NH high-latitude eruptions from other eruptions, as they require different scaling functions to relate

ice-sheet wide sulfate deposition rates to stratospheric aerosol loading (Gao et al., 2007). We defined NH high latitude eruption as eruptions that occurred at a latitude above 40° N. To identify the NH high latitude eruptions, we applied a Support Vector Machine learning classifier model (SVM – see methods section), that is being trained by the bipolar sulfate deposition of volcanic eruptions for which the eruption site is known. We applied 17 Holocene and 4 glacial volcanic eruptions of known origin (Table S6) to predict that 51 out of 85 bipolar

eruptions of unknown origin are likely to have occurred in the NH high latitudes (Fig. 5 and Fig. 6). The Gao et al. (2007) approach provides an estimate of the global stratospheric sulfate aerosol loading that is then linearly scaled to an atmospheric optical depth (Crowley and Unterman, 2013) and radiative forcing using the scaling factor of Hansen et al. (2005). The obtained radiative forcings are directly comparable to those obtained for the last 2500 years (Sigl et al., 2015).



Having estimated the volcanic forcing of the glacial and early Holocene eruptions, we can compare to the well-constrained eruptions that occurred over the last 2500 years (Sigl et al., 2015). In Table 2 we compared the number of eruptions with a climate forcing larger than Oruanui, Taupo, 25.5 ka; the unidentified eruption of 426 BC and Tambora, 1815 AD for the periods of 2.5-0 ka, and 60-9 ka. Note that the investigated period of the last glacial and the early Holocene covers some 43 ka (60-9 ka minus the section of no identified bipolar eruptions

16.5 to 24.5 ka). Among the 85 bipolar events investigated in the last glacial and early Holocene, three eruptions are found to be larger than Qruanui, Taupo, and 27 eruptions are found to be larger than the largest eruption of the past 2500 years.

Based on geological evidence, the occurrence of large volcanic eruptions was estimated in terms of their VEI (Volcanic explosivity index), which is based on the volume of ejected magma. The average occurrence rate of

VEI-7 (VEI-8) volcanoes discovered in the last 125 ka BP (2600 ka BP) is 0.3 (0.01) per millennium, respectively, based on databases of LaMEVE and GVP (Papale, 2018). From ice-core records, we cannot directly derive the VEI index, but the volcanic forcing estimated is likely to be another good indicator of volcanic eruption magnitude. The average rate of volcanoes in the 60-9 ka interval with a climate forcing larger than the 1815 AD Tambora eruption (VEI-7) is 1.60 per millennium (Table 2). In the same interval, we have 4

eruptions with a climate forcing larger or equal to the 25.47 ka Taupo Oruanui (VEI-8) eruption. Of those, the Icelandic 55.38 ka NAAZ II volcanic event is likely to be overestimated due to its proximity to Iceland (see section 4.4.) and it may not be of VEI-8 magnitude. Thus, we identify three VEI-8 sized eruptions occurring at a rate of 0.07 eruptions per millennium. Those ice-core based eruption frequency estimates are thus about 5.3 (7) times larger than those based on geological evidence for VEI-7 (VEI-8).

**4.4 Large and notable volcanic eruptions of the last glacial period (60-9 ka)**

The largest eruptions of the last glacial period and early Holocene are listed by the average climate forcing in Table 1. The highest ranking on the list is the Icelandic eruption associated with the NAAZ II occurring in GS-15.2 at 55.4 ka b2k (Gronvold et al., 1995; Zielinski et al., 1997). The sulfate deposition in Greenland from this eruption is enormous (almost 15 times higher than the deposition for Laki 1783 AD), but none or only very little

sulfate is deposited in Antarctica. With its close proximity to Greenland, it is therefore likely that a large fraction of the sulfate was transported to Greenland through the troposphere and thus the estimated climatic forcing of $-(99.7\text{-}228.0)$ W m$^{-2}$ is probably overestimated and should not be taken at face value. From the marine sediment records, the NAAZ II layer consists of different Icelandic events (Rutledal et al., 2020). For the





above reasons, the eruption should not be considered as the largest eruption of the last 60 ka in a global context.

The very wide range of sulfate deposition of 866.2-1435.6 kg km$^{-2}$ among the different Greenland ice-core records (Table S5) is probably due to a combination of 1) a true geographical variability in deposition in Greenland, 2) the quite significant and possibly inaccurate correction for layer thinning at great depth in the ice cores for this eruption, and 3) an unexplained difference related to the different analytical method applied (see results section).

The second largest event on the list occurred in GI-12 at 45.56 ka b2k and is identified in all Greenland and Antarctic ice cores, but the deposition at EDC is very low, most likely due to its low accumulation and the low time-resolution of the sulfate record. The eruption is of unknown origin, but based on its relative Greenland and Antarctic sulfate deposition, it occurred in the NH extratropical area and its climate forcing is estimated to the range of –(71.8-176.5) W m$^{-2}$.

Number three on the list of large eruptions occurred at 38.13 ka b2k; 100 years after the onset of GI-8 and 11 years before the occurrence of the Faroe Marine Ash Zone III (FMAZ III) tephra in Greenland (Davies et al., 2012). The eruption is detected in all ice cores (but there is a data gap for the WDC sulfur in this interval), and with very similar Greenland and Antarctic sulfate depositions this is most likely a low-latitude eruption with an estimated average climate forcing of -82.8 W m$^{-2}$, about 4 times that of Tambora (1815 AD).


The forth largest bipolar ice-sheet sulfate deposits are from the New Zealand Taupo, Oruanui, eruption that occurred in GS-3 at 25.46 ka b2k, verified by the presence of a tephra deposit (Dunbar et al., 2017). The estimated climate forcing of the eruption is in the range of –(33.2-118.1) W m$^{-2}$, about 2-3 times higher than Samalas 1257AD (-32.8 W m$^{-2}$) (Sigl et al., 2015), which caused large variability of atmospheric temperature of

global climate for several years (Lim et al., 2016), prolonged the drought in North American and leading to the famine (Herweijer et al., 2007). Despite its magnitude, the sulfate deposition of the eruption is undetected in the Greenland GISP2 ice cores with the lowest temporal sulfate-data resolution in this time period.

In GS-5.1 at 29.68 ka b2k there is a pair of volcanic eruptions separated by some 40 years, of which the younger event ranks 5 on the list of the largest eruptions. The estimated climatic forcing of this eruption is -65.5 W m$^{-2}$

and it has a clear NH signature in the bipolar sulfate distribution. There is no associated tephra deposit in the ice cores, but we propose the event originates from the Italian Campi Flegrei Y-3 eruption, being the only bipolar





volcanic eruption of that magnitude within a range of several millennia. The Y-3 eruption is independently Carbon-14 dated to 29.0 ka b2k (Albert et al., 2015).

Number 7 on the list of the largest eruptions occurred at 46.68 ka b2k in GI-12 and is the only eruption in this study apart from the 25.46 ka b2k Taupo, Oruanui, eruption that has a clear southern hemispheric origin. The Antarctic sulfate deposition of the eruption is approximately twice that in Greenland (Fig. 6) and the eruption has an estimated climatic forcing of -63.2 W m$^{-2}$ or about 0.8 times that of Oruanui.

Right after the onset of GS-9 at 39.92 ka b2k there is another pair of large eruptions separated by some 46 years. Those are ranking respectively number 15 (39,915 a b2k; -44.8 W m$^{-2}$) and number 38 (39,869 a b2k; -30.3 W m$^{-2}$) of the large eruptions listed in Table 1. Both eruptions have bipolar sulfate distributions suggesting a NH eruption. Because of their magnitude and their stratigraphic setting right at the onset of GS-9 these volcanic events are both possible candidates for the Italian Y-5 Campanian Ignimbrite eruption. There is no tephra evidence for this suggestion in the ice cores, but tephra from this eruption has been identified in the Black Sea in very similar stratigraphic setting and the eruption is independently dated by Ar/Ar to 39.9 ± 0.1 ka b2k (Giaccio et al., 2017).

In the late GI-1 right before the onset of GS-1 / Younger Dryas there is a quadruple of bipolar eruptions covering a period of 110 years (Svensson et al., 2020). The oldest (13,028 a b2k; -44.1 W m$^{-2}$) and the youngest (12,917 a b2k; -40.2 W m$^{-2}$) of those eruptions are number 17 and 19 on the list of large eruptions. Both eruptions are likely to have occurred in the NH at above 40°N. The 12,917 b2k eruption has been suggested as a candidate for the Laacher See eruption (LSE) that occurred in the East Eifel region in present-day Germany (Brauer et al., 1999; Baldini et al., 2018), but no tephra has been identified in the ice cores from this eruption. A recent publication, however, suggested that the LSE was around 130 years older (Reinig, et al., 2021) and is synchronous instead (within age uncertainty) with the oldest event of this quadruple of bipolar eruptions, but also with some minor sulfate signals.

The Vedde Ash layer (number 36) originates from a significant euption from Katla, Iceland with widespread tephra deposition in Greenland (Mortensen et al., 2005; Gronvold et al., 1995) and the North Atlantic region in the middle of GS-1 / Younger Dryas at 12.17 ka (Lane et al., 2012). Surprisingly, this Icelandic eruption not only deposited large amounts of sulfate in Greenland but also a much smaller amount in Antarctica, where it is identified in the WDC and EDML sulfate profiles suggesting that it had a large stratospheric injection. The volcanic forcing is estimated to about -30.9 W m$^{-2}$ taken with the reservation that some of the sulfate could have made it to Greenland through the troposphere.



**5 Conclusion**

We have employed three Greenland and three Antarctic sulfate/sulfur ice-core records to document the global
volcanic activity over the period 60-9 ka b2k. Detection of volcanic signals in the last glacial is challenging due

to extensive layer thinning of the ice cores with depth and a highly variable sulfate background level in the
Greenland ice cores across DO events, among other factors. Due to those challenges, we limited ourselves to
identify Greenland (Antarctica) eruptions with sulfate deposition larger than 20 (10) kg km$^{-2}$, which for
Greenland is about half the volcanic sulfate deposition of the Tambora 1815 AD and for Antarctica comparable
to the Pinatubo 1991 AD eruption.

With those restrictions we identified 1113 volcanic eruptions in Greenland and 740 volcanic eruptions in
Antarctica verifying the Northern Hemisphere (NH) to be the most volcanic active. Of those, 85 volcanic
eruptions had global impact (bipolar volcanos) as they were previously identified in Greenland and Antarctic
ice-core records. Compared to the past 2500 years, the ratio of bipolar volcanoes to total identified volcanic
events is very low in last glacial period, highlighting the challenges of confident synchronization over the glacial

period. Based on the hemispheric partitioning of sulfate deposition following well-known historical eruptions,
we determined if the bipolar volcanoes are likely to be situated above or below 40 ° N latitude. We then
estimated their climatic impact in terms of radiative forcing based on established methods. Throughout the
investigated period, we find that 69 volcanoes are larger than the Tambora 1815 AD eruption, and one unknown
volcano occurring at 45.56 ka b2k in the NH and one unknown volcano at 38.13 ka b2k in the low latitude or

Southern Hemisphere are larger than the Taupo, Oruanui, eruption occurring at 25.46 ka b2k in present day New
Zealand. The Icelandic NAAZ II eruption (55.38 ka b2k) has by far left the largest sulfate deposition in
Greenland, but due to its minor sulfate deposition in Antarctica, it is thought that only a fraction of the sulfate
was injected to the stratosphere. In general, we observe significantly higher occurrences of very large eruptions
(VEI 7 or larger) than estimated from the geological record, indicating that ice cores are more consistent than

geological evidence for volcanic signals detecting.

Overall, the frequency of volcanic eruptions per millennium is rather constant throughout the investigated period
and comparable to that of the most recent millennia. In agreement with previous studies, however, we find
elevated levels of volcanic activity in the NH during the deglacial period (16-9 ka b2k). In particular, many very
large eruptions occurred in this interval, quite likely associated with the redistribution of mass related to the melt

down of the major glacial ice sheets in this period. An apparent increase of Northern Hemispheric volcanic



activity in cold stadial periods as compared to milder interstadial periods is likely to be an artefact due to the highly variable nature of the Greenland sulfate signal of those periods.

Data availability: The high-resolution NGRIP CFA sulfate dataset is available as supplementary information for this publication. All other applied datasets are available elsewhere.

Author contributions: Initial idea was designed by A.S.. Datasets were collected by A.S. and J.M.L.. J.M.L. analysed the data and created the figures. J.M.L. and A.S. prepared the manuscript. All authors contributed to the discussion and commented on the manuscript.

Competing interests: The authors declare no competing interests.

Acknowledgements: A.S., C.H., J.P.S., B.M.V., S.O.R. and D.D.J. were supported by the IceFlow grant (grant
no. 00016572) from the Villum Foundation. J.M.L. is supported by a grant from the China Scholarship Council
(grant no. 201904910426).



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



Figure and table captions:

Table 1

The largest bipolar volcanic eruptions ranked by the volcanic aerosol forcing that is derived from the average Greenland and Antarctic sulfate depositions. The stratospheric aerosol loading was estimated with the same method as Gao et al. (2007) and the radiative volcanic aerosol forcing was established by scaling the atmospheric aerosol loading for Pinatubo 1991 (section 4.3). The 'Prediction of volcanic site' column is adapted from Table S5.

Table 2

The number of volcanic bipolar eruptions of comparable magnitude to those of Oruanui, Taupo (25.4 ka b2k), the largest eruption in last 2500 years (426 BC) and the Tambora 1815 AD eruptions by the climate forcing.

Figure 1

The average volcanic sulfate deposition in Greenland and Antarctica (60-9 ka; 2.5-0 ka adapted from Sigl et al., 2013, 2015). Blue line represents $\delta^{18}O$ of NGRIP core. Volcanic sulfate deposition of Greenland is larger than 1035 20 kg km$^{-2}$ (a, cyan bars) and Antarctic volcanic sulfate deposition is larger than 10 kg km$^{-2}$ (b, black bars). Bipolar volcanic eruptions are indicated with an 'x'.

Figure 2

Comparison of number of volcanoes per millennium grouped for different climate events by Greenland and Antarctic ice cores. The detected eruptions are classified into three size fractions, based on the 0.7 and 0.9 1040 quantiles of the volcanic sulfate deposition distribution, and the corresponding values of Greenland are 68 kg km$^{-2}$ and 140 kg km$^{-2}$ and for Antarctica the values are 25 kg km$^{-2}$ and 50 kg km$^{-2}$. (a) The number of volcanoes in last 2000 years in Antarctica is detected by 2-year sulfate resolution of in WDC core from Sigl et al., 2013. As only the WDC ice core is included in the above period, a scaling factor is applied to obtain the average sulfate deposition in Antarctica. For the DG, G, GI, GS periods the number of volcanoes per millennium is 1045 detected in all Antarctic ice cores with WDC sulfate records smoothed to 2 years resolution. (b) The number of volcanoes for the Holocene period is detected in the NEEM S1 core using a 2-year sulfate resolution with a rescaling factor for obtaining Greenland average sulfate deposition from Sigl et al., 2013. For the DG, G, GI, GS



periods, NGRIP is applied in 2-year resolution, and GISP2 is smoothed in 5-year resolution in the period 14-7 ka and for the period older than 14 ka b2k only larger eruptions are included).

Figure 3

Cumulative number and sulfate deposition of volcanoes with time for three classified groups of volcanoes for Greenland and Antarctica (60-9 ka). Volcanoes are classified into three groups: 20-68 kg km$^{-2}$, 68-140 kg km$^{-2}$, and >140 kg km$^{-2}$ for Greenland and 10-25 kg km$^{-2}$, 25-50 kg km$^{-2}$, and >50 kg km$^{-2}$ for Antarctica.

Figure 4

Number and sulfate flux of volcanoes per millennium with relatively similar data resolution (60-9 ka). Positive values represent Greenland volcanoes and negative values represent Antarctic volcanoes. Volcanic eruptions are grouped into three fractions based on the 0.7 and 0.9 quantiles of the volcanic sulfate deposition distribution, and the corresponding values of Greenland are 68 kg km$^{-2}$ and 140 kg km$^{-2}$ and for Antarctica the values are 25 kg km$^{-2}$ and 50 kg km$^{-2}$. The magnitude of volcanoes in the period of 60-9 ka is represented by the average
sulfate deposition of volcanoes in Greenland and in Antarctica. Volcanic list of Greenland is derived by smoothed sulfate record from GISP2 and NGRIP. For GISP2, the data was smooth with 5 years in the period of 14-9 ka and for the period older than 14 ka only large volcanic eruptions of GISP2 sulfate record is applied. NGRIP sulfate data has been smoothed to 2-year resolution. From 12-9 ka, number and sulfate deposition of volcanoes per millennium is based on GISP2 of Greenland ice core and EDML, EDC and WDC of Antarctica
ice core. For the latest period 911-1911 AD, 89-911 AD, volcanic sulfate deposition is defined from NEEM ice core of Greenland and WDC ice core of Antarctica (Sigl et al. 2013) and volcanic sulfate records of WDC and NEEM were detected with smoothed data of 2-year resolution. Volcanic list of Antarctica is detected with smoothed sulfate record of WDC in 2-year resolution.

Figure 5

Volcanic aerosol forcing of eruptions identified in both hemispheres. The size of the circle represents the strength of volcanic aerosol forcing.

Figure 6



Relationship between Greenland and Antarctic volcanic sulfate deposition of bipolar eruptions classified by Support Vector Machine (SVM) model. Red circles stand for Greenland volcanic sulfate deposition and blue

circles stand for Antarctic sulfate deposition. The size of the circle stands for the strength of volcanic deposition and volcanic aerosol forcing.





Table 1

| Rank | Age (b2k) | Greenland volc. depo. (kg km$^{-2}$) | Antarctica volc. depo. (kg km$^{-2}$) | Stratospheric aerosol loading (Tg) | Global climate forcing (-W m$^{-2}$) | Prediction of volcanic site |
|---|---|---|---|---|---|---|
| 1 | 55383 | 1117.4 | 16.5 | 653.4 | 130.7 | NHHL |
| 2 | 45555 | 652.6 | 172.7 | 544.7 | 108.9 | NHHL |
| 3 | 38133 | 209.4 | 204.5 | 413.9 | 82.8 | LL or SH |
| 4 | 25460 | 189.5 | 192.8 | 382.3 | 76.5 | LL or SH |
| 5 | 52302 | 293.0 | 185.4 | 352.4 | 70.5 | NHHL |
| 6 | 29680 | 333.3 | 137.6 | 327.5 | 65.5 | NHHL |
| 7 | 46683 | 105.1 | 211.1 | 316.2 | 63.2 | LL or SH |
| 8 | 41144 | 410.0 | 81.8 | 315.5 | 63.1 | NHHL |
| 9 | 49065 | 164.9 | 142.3 | 307.2 | 61.4 | LL or SH |
| 10 | 42037 | 327.7 | 119.8 | 306.6 | 61.3 | NHHL |
| 11 | 44763 | 169.7 | 171.0 | 267.7 | 53.5 | NHHL |
| 12 | 32032 | 210.8 | 145.6 | 265.8 | 53.2 | NHHL |
| 13 | 10481 | 312.5 | 56.7 | 234.9 | 47.0 | NHHL |
| 14 | 25759 | 214.8 | 110.6 | 233.1 | 46.6 | NHHL |
| 15 | 39915 | 316.0 | 44.0 | 224.1 | 44.8 | NHHL |
| 16 | 34718 | 301.0 | 52.5 | 224.1 | 44.8 | NHHL |
| 17 | 13028 | 272.8 | 65.0 | 220.5 | 44.1 | NHHL |
| 18 | 11305 | 297.8 | 45.6 | 215.3 | 43.1 | NHHL |
| 19 | 12917 | 248.1 | 59.4 | 200.8 | 40.2 | NHHL |
| 20 | 47023 | 255.4 | 52.3 | 197.9 | 39.6 | NHHL |
| 21 | 16469 | 102.9 | 89.8 | 192.7 | 38.5 | LL or SH |
| 22 | 14761 | 307.8 | 10.7 | 186.2 | 37.2 | NHHL |
| 23 | 28942 | 88.4 | 97.5 | 185.9 | 37.2 | LL or SH |
| 24 | 42250 | 183.0 | 80.9 | 185.2 | 37.0 | NHHL |
| 25 | 35556 | 236.4 | 45.4 | 180.1 | 36.0 | NHHL |
| 26 | 44507 | 98.6 | 80.3 | 178.9 | 35.8 | LL or SH |
| 27 | 40183 | 211.5 | 47.5 | 168.1 | 33.6 | NHHL |
| 28 | 58182 | 156.9 | 78.6 | 168.1 | 33.6 | NHHL |
| 29 | 27797 | 263.9 | 17.6 | 168.0 | 33.6 | NHHL |
| 30 | 46116 | 66.5 | 99.4 | 165.9 | 33.2 | LL or SH |
| 31 | 42658 | 133.1 | 85.6 | 161.5 | 32.3 | NHHL |
| 32 | 24669 | 172.9 | 62.1 | 160.7 | 32.1 | NHHL |
| 33 | 15559 | 84.0 | 72.2 | 156.2 | 31.2 | LL or SH |
| 34 | 30244 | 208.7 | 36.3 | 155.2 | 31.0 | NHHL |
| 35 | 29722 | 206.1 | 36.8 | 154.3 | 30.9 | NHHL |
| 36 | 12170 | 248.6 | 11.2 | 153.0 | 30.6 | NHHL |
| 37 | 16334 | 187.5 | 45.8 | 152.7 | 30.5 | NHHL |
| 38 | 39869 | 158.5 | 61.2 | 151.6 | 30.3 | NHHL |
| 39 | 38366 | 227.6 | 16.0 | 145.7 | 29.1 | NHHL |
| 40 | 59647 | 129.0 | 72.0 | 145.6 | 29.1 | NHHL |
| 41 | 37965 | 88.7 | 56.0 | 144.7 | 28.9 | LL or SH |
| 42 | 53259 | 146.1 | 56.8 | 140.0 | 28.0 | NHHL |
| 43 | 14966 | 81.7 | 50.5 | 132.1 | 26.4 | LL or SH |
| 44 | 57051 | 69.6 | 57.6 | 127.2 | 25.4 | LL or SH |
| 45 | 37277 | 149.3 | 35.3 | 120.4 | 24.1 | NHHL |




Table 2

| | Absolute number of eruptions 9-16.5 ka | | Eruptions per millennium 9-16.5 ka | |
| | 0-2.5 ka | 24.5-60 ka | 0-2.5 ka | 24.5-60 ka |
|---|---|---|---|---|
| > 75 W m$^{-2}$ (Oruanui, Taupo) | 0 | 3 | 0 | 0.07 |
| > 35 W m$^{-2}$ (EU 426 BC) | 1 | 27 | 0.4 | 0.63 |
| > 17 W m$^{-2}$ (Tambora) | 6 | 69 | 2.4 | 1.60 |




Figure 1







Figure 2


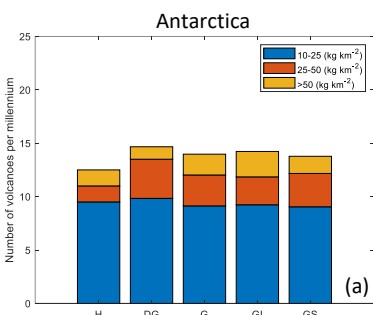

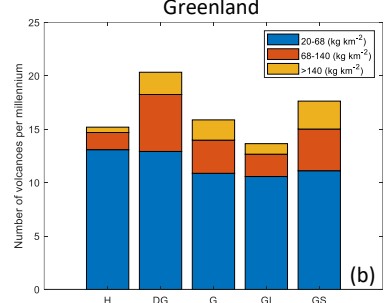

H = Holocene period (2-0 ka b2k)
DG = Deglacial period (21-9 ka b2k)
G = Glacial period (60-21 ka b2k)
GI = Glacial interstadials of G
GS = Glacial stadials of G

Figure 3

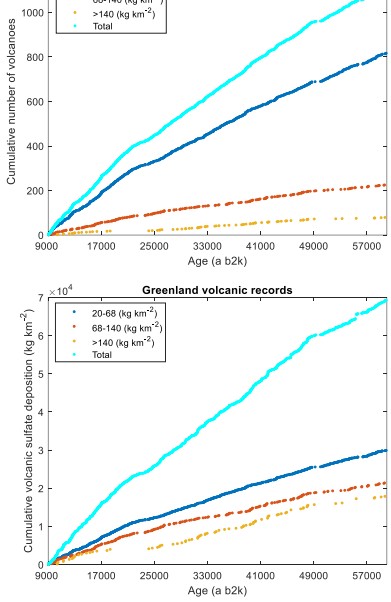

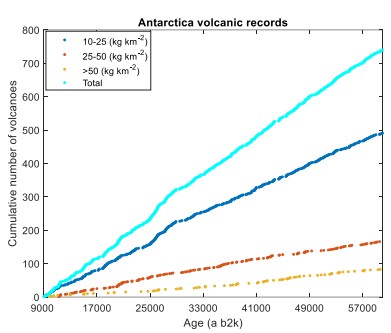

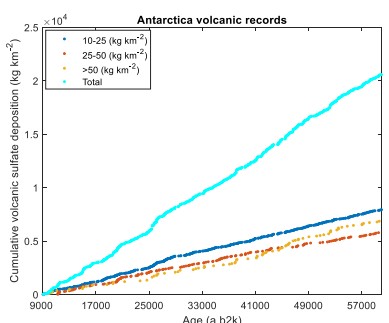

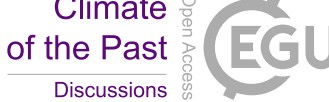

Figure 4

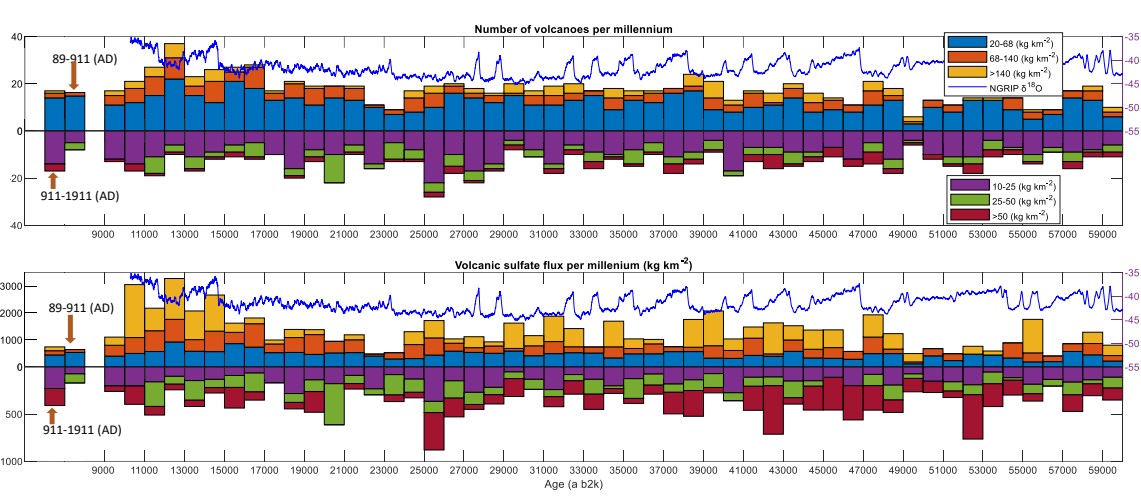

Figure 5

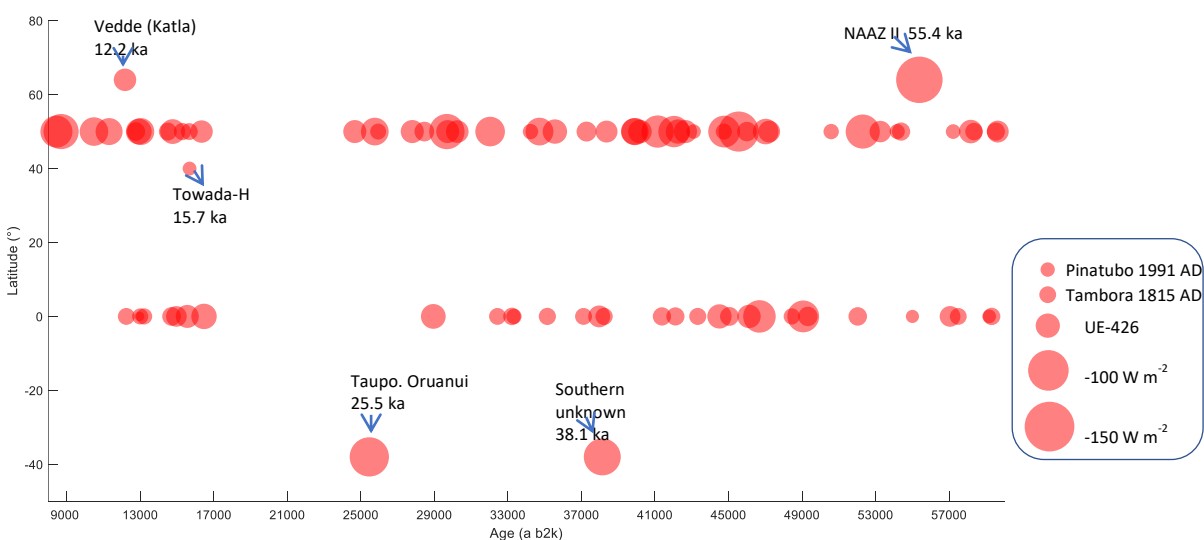





Figure 6

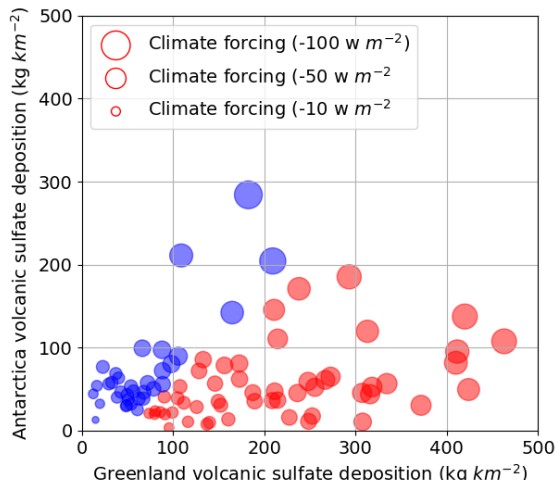