# Peer review of "Magnitude, frequency and climate forcing of global volcanism during the last glacial period as seen in Greenland and Antarctic ice cores (60-9 ka)"

_Climate of the Past, 2021_

## Author Comment (AC1)

Response to the comments from Alan Robock.

This looks like an excellent paper, but I noticed a few things that could be improved.

We thank you for taking time to improve the manuscript. We will revise the paper according to the suggestions.

In the abstract, change "The frequency of eruptions larger than the typical VEI-7 (VEI-8) eruption by the comparison of sulfur emission strength is found to be 5.3 (7) times higher than estimated from geological evidence." to "The frequency of eruptions with sulfur emissions larger than the typical VEI-7 eruption is found to be 5.3 times higher than estimated from geological evidence, and for VEI-8 eruptions it is 7 times higher." Using parentheses to save space only serves to confuse and make it difficult to read.

This sentence has been deleted in the abstract and reformulated in the section 4.3. It now says: 'The average rate of volcanoes in the 60-9 ka interval with a climate forcing larger than the 1815 AD Tambora eruption (VEI-7) is 1.60 per millennium (Table 1). In the same interval, we have 4 eruptions with a climate forcing larger or equal to the 25.32 ka BP1950 Taupo Oruanui (VEI-8) eruption. '

Furthermore, the sentence is awkwardly constructed. But VEI is not an index of sulfur emission. Why use VEI at all when discussing the impacts of volcanic eruptions on climate?

Thank you for the suggestion. Indeed, VEI is an index of the volume of ejected magma. We have included a sentence in section 4.3 'From ice-core records, we cannot directly derive the VEI index, but the estimated volcanic forcing is likely to be another good indicator of volcanic eruption magnitude. The average rate of volcanoes in the 60-9 ka interval with a climate forcing larger than the 1815 AD Tambora eruption (VEI-7) is 1.60 per millennium...'.

It would be much easier for reviewers if you put the table and figure captions on the same page as the tables and figures.

Indeed, we have included the table and figure captions on the same page as the figure/table.

Table 1 uses acronyms that are not defined. What are NHHL, SH, or LL?

Those are now defined in the caption of Table 1.

Table 2 needs to be corrected. Radiative forcing from volcanic eruptions is negative. For example, for those with forcing larger than Tambora, they are for forcing < –17 W m-2.

This has been corrected.

The notation in Fig. 6 needs correction. The correct unit for radiative forcing is W m-2, with W capitalized, and m not in italics. Italics are for variables, and not for units. Similarly, km should not be in italics.

Thank you for the clarification. This has been corrected.

Review by Alan Robock

---

## Author Comment (AC2)

Response to the comments from Chaochao Gao.

We thank the reviewer for taking the time to consider this long manuscript in details. Detailed replies are given in the following.

Reply to specific comments:

1. 'Section 2.2, please explain briefly the choice of different filter length, i.e., 45 yr for the Antarctic cores and NGRIP while 181yr for GISP2 & NEEM.'

   The filter length choice is empirical, depending on the depth resolution of the sulfate records for the individual cores. The criteria are to ensure that the sulfate background is smoothed for non-volcanic high-frequency spikes and at the same time preserves the abrupt changes across climate transitions to the highest degree possible.

2. 'Also in section 2.2, it is a bit confusing about which parameter was used to measure the volcanic signals, the depth or the yr. Based on the text, the background was filtered by window length indicated by year, while the duration of the volcanic signal was indicated by depth.'

   We use both the original depth scale and an interpolated time scale. The filtering and the event duration are using the time scale, but in order to preserve the maximum resolution we do the integration of the sulfate spike using the original dataset on a depth scale.

3. 'Section 2.4 please provide some description of how the manual correction was performed. For example, what are the resolutions for the ECM and DEP records? How many or what percentage of the signals have been corrected. It would also be great to give an example of how the correction was done'.

   A manual correction was done for multiple volcanic sulfate peaks that are merged into one peak by separating neighbouring peaks according to the corresponding high-resolution DEP or ECM peaks. We added some explanation in the text line 229-233: 'As the depth resolution of the sulfate records for the GISP2 and NEEM cores is relatively low (Fig. S2 (a)), adjacent acidity peaks may be merged into falsely large acidity spikes. We made a manual correction for this effect by comparing to the corresponding higher-resolution ECM (NEEM in 10 mm and GISP2 in 1-5 mm resolution) and DEP (NEEM in 5mm resolution) records of the same core and split falsely large peaks according to the associated ECM or DEP peaks for the top 50 largest events and removed the peaks for smaller events. The specific correction for individual volcanic signals is indicated in Table S3. Twenty volcanic events were corrected for NEEM and 14 volcanic signals were corrected for the GISP2 core.

   One example of this correction is shown in the following figure. The volcanic sulfate peak in NEEM at 1461.3-1461.6 m depth includes three ECM and DEP peaks. We split this suflate peak into three volcanic sulfate signals and assign the sulfate deposition values according to the proportion of DEP or ECM peak areas. This approach assumes that sulfate is the dominant acid contributing to the electric signals.

[Figure]

4. 'A sulfate deposition of 20 kg km$^{-2}$ corresponds to half the Greenland deposition from the 1815 AD Tambora eruption, thus refers to quite large events in terms of total sulfur injections into the atmosphere." Please explain briefly how was the 20 kg km$^{-2}$ (and also the 10 kg km$^{-2}$ for Antarctic) cut-off line estimated. And was the 40 kg km$^{-2}$ 1815 AD Tambora deposition corresponding to the average deposition from the three Greenland cores?'

   The 20 kg km$^{-2}$ and 10 kg km$^{-2}$ cut-offs were applied because it becomes difficult to distinguish the sulfate background variability from volcanic eruptions for smaller events. In Greenland the variability of the background is much higher than that in Antarctica, therefore the cut-off needs to be higher. Another reason for applying the cut-offs is to obtain a dataset that is consistent through the whole

investigate period. Without using a cut-off to identify volcanic events we would detect more smaller events in the most recent part of the records where the temporal resolution is higher. By making a conservative cutoff for the entire profile we homogenize the dataset. The volcanic sulfate deposition 40 kg km$^{-2}$ for Tambora 1815 AD is that obtained by Sigl et al., 2015 and is the average of NEEM-2011-S1 and NGRIP.

5. 'Section 2.6 Please provide more details on the SVM model. For example, what are the requirements, the pro. and cons of the model in this particular application. What validation had been done on the model performance? For example, taken one of the 21 eruption signals used to train the model out from the analysis, could the model accurately simulate its location?'

More description for this model was added to the text in Section 2.6 and the caption of Fig. S4. 'The volcanic sulfate deposition in Greenland and Antarctica shows a distribution pattern related to the latitudinal band of the eruption site (Fig. 1) (Marshall et al., 2019). To estimate the latitudinal band of bipolar volcanic eruptions of unknown origin, we applied the Support Vector Machine (SVM) classification model of Hastie et al. (2009) and Vapnik (1998). The classification model is based on a kernel function generation and logistic regression. The model was trained using 21 eruptions for which the eruption site is known from tephra deposits in the ice (Table S6). The input values for each eruption to the model are the average Greenland sulfate deposition, the average Antarctic sulfate deposition and the latitudinal band (above 40°N, 40°N-40°S, or below 40°S) of the eruption site. The cross-validation used for tuning the algorithm is 10-fold partition for each evolution. For an optimal classification, a maximum-margin hyperplane was used to separate two classes. The kernel scale and box constraints were chosen for the model and a Bayesian optimization was used to optimize the above two parameters to yield the best classification model (Fig. S4) (more detailed descriptions are in the Hastie et al. (2009), page 17). The bipolar eruptions of unknown origin were predicted into two latitudinal bands – above 40°N (NHHL) and below 40°N (LL or SH) (Table S5) based on the trained model. Due to the low number of known volcanoes erupted in the high latitudes of the Southern Hemisphere, the method does not allow unambiguous identification of eruptions potentially located in this region.'.

The caption of Figure S4 is now: '(a) The samples (trained + predicted) are classified by latitudinal bands: above 40°N (NHHL) in red '+', below 40°N (LL or SS) in green '*'. The support vectors, that are shown as circles close to the hyperplane, are applied to tune the hyperparameters. (b, c) Bayesian optimization of the model with two parameters (kernel scale and box constraint) yields the best classification model.'.

6. 'Ln 305-307, the comparison between IC and CFA records in Fig S2e needs further demonstration. For example, what is the exact meaning of "very large uncertainties"? What is the implication of the uncertainties on the interpretation of the "face value"?'

In line 305-307 and Fig. S3 (e), 'the very large uncertainty' estimation is based on comparing the same volcanic sulfate peak in NGRIP ice core as derived by the CFA and IC analytical methods. For this comparison, we can exclude uncertainties related to the ice flow model (layer thinning), as they are obtained from the same core. The average sulfate deposition measured by CFA is around 20% higher than that obtained by IC. This difference may be caused by the different analytical techniques and by the different sample resolutions between CFA and IC.

'Section 3.3 Please explain why borrow the bipolar eruptions from the previous studies, rather than estimate a list using the results from this study.'

This may be a question related to section 4.3. We compare the magnitude estimates of the glacial eruptions to already published values for volcanic eruptions of the most recent 2500 years in order to set them in a historical context. We are not recalculating the magnitudes of the eruptions of the last 2500 years because the applied methods are similar and because we do not apply the exact same sulfate records for the last glacial as for the historical period. For example, the high-resolution NGRIP CFA dataset only covers the glacial period and is not available for the Holocene.

7. 'Ln 532-534 In my understanding, the VEI list is a discrete (i.e., it is not a complete but continuously evolving) reconstruction of historical volcanism based on geological investigation. So, I am not sure it is appropriate to directly compare the event frequency from the ice-core-based reconstruction (which is assumed to be continuous) with that from geological investigation.'

Indeed, the VEI scale is based on the volume of ejected magma combined with other parameters such as the height of ejected ash cloud (Pyle, D., 2000) and not including the volcanic sulfur emission strength. However, we want to have an impression of the magnitude and frequency of volcanic sulfur emissions from the ice-core-based reconstruction. So, we chose the well-known eruptions (Table S6) and compared their volcanic sulfur emission strength (the stratospheric aerosol loading) to the VEI scale as shown in the following figure. The figure shows that to first order there exists a positive relationship between the VEI scale and the volcanic sulfur strength.

Caption of the below figure: Comparison of stratospheric aerosol loading to the VEI of the same volcanic events. The red line is the median value (values are shown in the upper part) and stars represent the average values. The box lines represent 25 and 75 percentiles. 5 percent outliers are indicated with '+'.

[Figure]

We have added the following discussion in section 4.3 'The above comparison rests on the observation that there exists a positive relationship between the volume of ejected magma and sulfur emission gas for a volcanic eruption.

8. Section 4.4 Please explain why do some events have forcing estimation in a range (for example, the #2 largest signal has forcing ranging from 17.8 to 176.5 W/m2), while others have finite forcing estimation (for example, the #3 largest signal has a forcing estimation of 82.8 W/m2)? If it was due to the number of ice cores available for signal extraction, this should be clarified.

Yes, that is correct, when we provide a radiative forcing range this means that the eruptions have been detected in several cores. We now state in Table 2 'Number of ice cores' refers to the number of ice cores in Greenland and Antarctica in which the volcanic sulfate signal has been detected.'. This clarification is also added in the section 4.4. 'The largest eruptions of the last glacial period and early Holocene are listed by the average climate forcing in Table 2'.

9. Is there any reason why the authors choose Tambora & Samalas for the magnitude comparison of #3 and #4 events, respectively?

The comparison to Tambora 1815 AD is because it is a well known and well-studied tropical VEI-7 eruption, and Samalas 1257 AD is the largest tropical volcanic eruption for which the source is known from the last 2500 years.

10. Please add captions for the supplementary figures.

The captions have been added to the supplementary figures.

Reference:

Pyle, D. Sizes of volcanic eruptions in: Encyclopedia of Volcanoes (ed. Sigurdsson, H.), 263–269 (Academic, San Diego, 2000).

---

## Author Comment (AC3)

Response to the comments from Eric Wolff.

'This paper represents a huge amount of work to identify the deposition of volcanic sulfate in the polar regions and to try and draw conclusions about the occurrence of volcanism over the last 60 kyr. I really applaud the effort, and the attempt to draw large-scale conclusions from it. I have a lot of relatively minor comments on the paper (which put it somewhere between minor and major revision), and I think the authors could have made some better choices in the way they treated the data. However, I accept that they made mainly reasonable choices and so I do not propose to insist on any significant reanalysis of the data – just in places an extra sentence is needed to discuss the choices made. I found some of the messages that end up in the abstract too strong given the nature of the data and I will comment on those in the text, expecting the authors also to address them in the abstract.'

We thank the reviewer for taking time to provide constructive comments to improve this work. We will revise the manuscript accordingly.

'A final point is that the paper is extremely hard to follow – while I appreciate the need for a lot of supplementary material, the fact that the figures in the main text and the supplement seem to be called in almost random order is very unhelpful, and I suggest the authors renumber the figures to provide a more logical flow.'

We have updated the order of figures and tables in the text to follow the flow of the text.

'Before I give a detailed set of comments, I should add that (as I think several of the authors are aware) I have been involved in a similar exercise, using only the EDC ice core, but for a period of 200 kyr. This has been presented a few times and has been (in a paper involving also terrestrial and marine data) under review for a considerable time. My comments should therefore be taken with the knowledge that I have used slightly different methods and criteria but am not seeking to impose the same methodology on the authors here.'

Thanks for sharing this. We look forward to see the exciting 200 kyr profile being published.

Detailed comments:

'The paper will need a thorough copy edit as the English at times is awkward and occasionally hard to follow.'

We went through the text and hope we have succeeded in making it more readable.

'Line 165. I think the authors have misunderstood the MSA correction. This had to be done for WAIS Divide because ICPMS measures elemental S, and therefore the values obtained are (sulfate + MSA). This is not the case for any of the FIC/IC methods, where sulfate itself is measured, so no correction would be required and no correction should have been made for any other sites. Please clarify this in your text.'

Thank you for this clarification. We completely agree and we have removed the discussion of MSA, except mentioning that we cannot do the correction for WDC in the glacial period.

'Line 188. I found myself a little confused about the volcanic detection threshold. I understand the idea behind the use of RRM and RMAD. But then in the end you just use a threshold of 10 or 20 kg km$^{-2}$ so I am not sure what purpose the statistical threshold serves.'

Indeed, we do the volcanic detection in two steps. First step being the 'classical' RRM and RMAD approach that works well in the Holocene and a second step being a reasonable and bold cut-off. Ideally, the second step should not be necessary, but we did it in order to obtain a more homogenous volcanic list throughout the investigated period. With all of the variation in both climate and data resolution, we found it safest to take this approach. We did experiments with different cut-off values and the applied values are chosen conservatively to avoid including non-volcanic sulfate spikes that may not be discarded in the 'classical' approach in particular during stadial periods.

'Equation 1: please be careful to define the units when describing this equation. I assume concentration is in ppb or ng/g (as in the figures), and D is in m ice equivalent (this is important as if you had included the top 100

m where density is less than that of ice, then the equation is not correct if in real depth), and 0.917 is in g/cm$^3$. By fluke, after cancelling factors of 10 to change the units this does indeed work out as kg km$^{-2}$, but the reader needs that to be stated.'

The units for the equation have been added in the text.

'Line 209. Please also be careful to define what T is, ie layer thickness/original layer thickness. It's confusing otherwise because you talk next about a 60% reduction of the layer thickness which is equivalent to a T of 0.4. Perhaps that is the best way to deal with it, "60% reduction of the layer thickness (T=0.4)".'

This has been clarified in the text. It now says: 'T is the layer thinning (the ratio between the layer thickness in the ice core and the original layer thickness). The layer thinning has been calculated by site-specific ice-flow models (thinning function) that we applied here (Table S1 and Fig. S2 (c)). During the last glacial period, the layer thinning can be significant. For the Greenland cores, the thinning rate ranges from a 60% reduction of the original layer thickness (T=0.4) at 12 ka b2k to as much as 90% (T=0.1) at 60 ka b2k (Table S1). For Antarctica, the thinning rate is most significant for the high-accumulation WDC core, where it approaches 95% (T=0.05) at 60 ka b2k (Fudge et al., 2016; Buizert et al., 2015), whereas the EDC core exhibits a modest thinning of 30% (T=0.7) at 60 ka b2k (Fig. S2 (c)).'

'Supp table S1: Please explain what the column "Name" is: I assume it's the age model used. But I'm surprised because the standard for a 60ka period is GICC05. I appreciate you may have to use a model to derive a smooth acc rate, but then you need to explain the relationship between ss09seabm1 and GICC05 as few of your readers will have heard of the former.'

The column 'Name' in the supplement Table S1 is the name of the ice-flow model of each core. An explaination for 'ss09seabm1' is added to the caption of Table S1 as follows: 'ss09seabm1 is the model timescale which include the layer thinning rate.'

'Fig S1 and caption. Please clarify that in part c the y-axis "thinning function" is equivalent to "T" in the text as I don't think you ever explain that in the text. Also in the caption, what does "thinning file" mean? The second sentence of the caption doesn't make sense, please re-word.'

The 'thinning function' has been clarified as follows: 'T is the layer thinning (the ratio between the original layer thickness and the layer thickness in the ice core). The layer thinning has been calculated by site-specific ice-flow models (thinning function) that we applied here (Table S1 and Fig. S2).'.

In the caption, 'thinning file' is now changed to 'thinning profile'. The second sentence of the caption is re-worded to 'A simple linear fit is applied to the WDC thinning profile below 2800m depth, labelled as WDC (combined), as the gas derived thinning profile has potentially unrealistic wiggles (Buizert et al., 2015).'.

'Line 220. This is wrong. The volcanic sulfate flux calculation does not assume anything about wet or dry deposition: it is simply the product of concentration and snow acc rate. It's true that the flux is affected by whether there is significant wet deposition in addition to the dry deposition and this is a point worth making, but the flux as calculated is definitely correct.'

This has been reworded. We edited the passage as follows: 'For high snow accumulation sites, such as those in Greenland and coastal Antarctica, it is quite likely that wet deposition is dominating the sulfate deposition at present day (Kreutz et al., 2000; Schupbach et al., 2018). During the last glacial period, dry deposition may have played a more important role, giving rise to a potential bias of our glacial sulfate deposition estimates. We do not attempt to make any corrections for this effect.'

'Table S4, column O, kg not Kg please.'

Corrected.

'Table S5. Why is N/A written for WD2014 ages below 30 ka. WD2014 extends from the surface to 68ka even if it changes from layer counted to methane-tied at 31 ka.'

Thanks for the suggestion. The WD2014 ages have been included throughout.

'Tables S4 and S5: Are the values given for individual sites before or after the rescaling of Antarctic concentrations. I think before – if not then it's hard to understand the averages you give for the Antarctic. Please clarify this in the captions, ie that the individual values are as measured while the average is after rescaling.'

This has been clarified in the supplementary caption of Table S4-S5 as follows: ''Deposition' is the sulfate deposition for individual events after correcting the layer thinning but without applying a scale factor to obtain the average Antarctic deposition. When an event is detected in three cores, the 'Average deposition' is the simple mean of the volcanic sulfate deposition at the three sites. When an eruption is detected in one or two cores, a scale factor is applied to obtain the averaged Antarctic deposition (see 'Methods 2.5').'.

'Line 241. What matters is of course not the value of the sulfate background but its variability. I think this is what you mean but from this wording it isn't quite clear. I think you need to explain your 10 and 20 kg m-2 criteria better – how is it derived? In my own work we estimated the "negative" peaks to understand how big a deviation from the median could be generated in a given section from noise alone, and then we used that to establish a threshold (which was 20 kg km$^{-2}$ for the 200 ka record). So I think your thresholds are probably fine, but could be explained more clearly.'

We have two main concerns: One is the inconsistent resolution of sulfate records over the investigated period due to the high-level layer thinning in the deep part of ice core. The other one is the cold Greenland periods (stadials), where the sulfate background variability is high and numerous short-lived sulfate spikes (<3 yr duration) get above the detection limit in the 'classical' volcanic detection scheme, in particular for the high-resolution NGRIP CFA profile. This is demonstrated in the figure below where the NGRIP CFA sulfate profile is exposed to different degrees of smoothing of 1, 2, 3 and 5 years, respectively. With the 1 and 2 year smoothing a large number of small peaks are detected, some of which may be of volcanic origin, but most likely a large fraction is related to gypsm or other sources of non-volcanic origin (as discussed in section 4.1). For Greenland, the detection of sulfate depositions larger than 20 kg km$^{-2}$ is rather insensitive to the degree of smoothing and we therefore chose this value for the cutoff. For Antarctica, the background variability is much lower and we find that 10 kg km$^{-2}$ is a 'safe' cutoff.

[Figure]

Caption of the above figure: The two plots in the upper panel show examples of background signal determination and volcanic peak detection for the NGRIP sulfate record as smoothed to 3-year and 5-year

resolution, respectively. The four plots in the lower panel are the number volcanoes detected in 1-year, 2-year, 3-year, and 5-year resolution for the entire NGRIP sulfate record.

'Line 255. The Antarctic plots shown in Fig S2 are really not impressive with very small r^2. They don't seem at all a good basis for the rescaling you do. While I accept there may be higher fluxes for a given volcano at WD because it receives more wet deposition (this is the only reason the accumulation rate is relevant), the values you derive here have a huge uncertainty which you have not apparently propagated into your 26% error estimate (line 265). We know there is a huge uncertainty anyway because of local variability in deposition (as shown eg by Gautier et al (2016). I suggest you do an alternative error estimate where you use the rescaled data from the 123 Antarctic eruptions where you have 3 sites and find the average 1-sigma and/or 2-sigma between the 3 sites. This would give you an alternative way to estimate the uncertainty in the values you end up with. 26% is certainly way too small an error in the cases where you have only 1 site.'

Indeed, we agree that the linear regression among three Greenland and three Antarctic ice cores is not a perfect approach for obtaining the rescaling. We used the error of propagation method to estimate the uncertainty of the sulfate deposition for individual volcanic events, considering the sulfate diffusion, the layer thinning, the uncertainty of the analytical method, and the data resolution. For EDC and EDML, we now include an additional uncertainty due to the demonstrated high local variability in sulfate deposition. The uncertainty of the local variability is 29% according to Gautier et al. (2016). Thus, the estimated error of the sulfate deposition for EDC and EDML is now up to 40%. We applied the method 'the average 1-sigma and/or 2-sigma between the 3 sites' to estimate the average uncertainty for the rescaled volcanic sulfate deposition from 124 events, where we have sulfate deposition at three sites. When we have less than three sulfate deposition values, we use the maximum uncertainty of the sulfate deposition of the individual cores.

We revised the wording of the caption for Table S4 and the Lines 268-273 'For EDC and EDML, the low accumulation leads to a large local variability in sulfate deposition of up to 29% (Gautier et al., 2016), so for those cores we arrive at an combined error estimate of 40% for individual deposition events. When there are sulfate signals from three ice cores in one hemisphere, we used the standard deviation of the rescaled volcanic sulfate depositions for all ice cores to estimate the uncertainty for the average area volcanic sulfate deposition. When there is a common signal in fewer than three cores, we use the maximum uncertainty of the volcanic sulfate deposition from the individual cores.'.

'Line 268. I am familiar with the paper by Marshall (not Martshall) but I don't know what you mean by "The volcanic sulfate deposition in Greenland and Antarctica shows a distribution pattern (Table S5 and Table S6), similar to that derived from the aerosol-climate modeling of volcanoes over past 2500 years". I can believe that you are assuming they show such a distribution and that this is how you decide on the latitude, but you have no evidence Section 2.6 and Fig S3. I apologise for my lack of technical knowledge but this section is not comprehensible to someone coming at it new. I think you need to explain what you are trying to achieve here (which I think is to use the Greenland/Antarctic ratio of known eruptions to classify the latitude of unknown bipolar eruptions). Then please try to explain SVM better. To anyone not in the know Fig S3 b and c cannot be understood.'

This paragraph has been revised as follows: 'The volcanic sulfate deposition in Greenland and Antarctica shows a distribution pattern related to the latitudinal band of the eruption site (Fig. 1) (Marshall et al., 2019). To estimate the latitudinal band of bipolar volcanic eruptions of unknown origin, we applied the Support Vector Machine (SVM) classification model of Hastie et al. (2009) and Vapnik (1998). The classification model is based on a kernel function generation and logistic regression. The model was trained using 21 eruptions for which the eruption site is known from tephra deposits in the ice (Table S6). The input values for each eruption to the model are the average Greenland sulfate deposition, the average Antarctic sulfate deposition and the latitudinal band (above 40°N, 40°N-40°S, or below 40°S) of the eruption site. The cross-validation used for tuning the algorithm is 10-fold partition for each evolution. For an optimal classification, a maximum-margin hyperplane was used to separate two classes. The kernel scale and box constraints were chosen for the model and a Bayesian optimization was used to optimize the above two parameters to yield the best classification model (Fig. S4) (more detailed descriptions are in the Hastie et al. (2009), page 17). The bipolar eruptions of unknown origin were predicted into two latitudinal bands – above 40°N (NHHL) and below 40°N (LL or SH) (Table S5) based on the trained model. Due to the low number of known volcanoes erupted in the high latitudes of the Southern Hemisphere, the method does not allow unambiguous identification of eruptions potentially located in this region.'.

The support vector machine learning model provides a statistical approach to predict the erupted latitudinal pattern for the past unknown volcanic eruptions.

'Line 282, I think you mean Fig S10, not S9.'

It has been corrected to Fig. S10.

General comment: this reminded me that the calling of figures, especially supplementary ones, in apparently random order is really confusing. Please sort this out.

This has been sorted.

'Line 360. I don't really understand your statement that "the layer thinning becomes stronger which makes it increasingly difficult to detect smaller eruptions signals". If that is really so then your threshold of 10 or 20 kg km$^{-2}$ is too low. The whole point of it should be to ensure that detection is the same throughout.'

Indeed, the threshold or cut-off at 10 or 20 kg km$^{-2}$ does the main part in homogenizing of the dataset over the investigated period. However, due to the very high resolution of the NGRIP CFA sulfate (1 mm resolution) and the WDC CFA sulfur (1 cm resolution), in order to discuss the frequency of events over time we also do a homogenization of those datasets by smoothing them to their 'effective' resolution that we deduce from their power spectra (Figure S6). We do not make this smoothing at an earlier stage because that would make us unable to detect smaller volcanic events for example during the deglacial period. It is only when we want to study the frequency of eruptions across the entire 51 ka period that we smoothen the high-resolution records to their lowest resolution during the cold periods early in the investigated period.

''Section 3.4. The previous comment raises a more general comment on this section. It's really obvious that you shouldn't use cores where the resolution is worse than the expected peak width. However, it's difficult to know what that means: in the case of EDC, if thinning were the only thing happening then it would be hopeless to detect eruptions reliably in the early parts of the record with resolution 5 years. However, it turns out that the diffusion at Dome C keeps the peak width quite constant with age, and therefore allows resolution of peaks where the age resolution is 5 years (because the peaks at this depth have diffused to 10 or 20 years wide). However, I'd be very surprised if you can expect to resolve most volcanoes, even large ones, in GISP2 with the stated resolution in Table S1. For NEEM, given the variable depth resolution, I suspect the age resolution is normally much better than 10 years (I suggest checking this again), but if it were 10 years again I think detection would be hopeless. I think you need a somewhat deeper discussion here, and also to consider whether it is worthwhile to contaminate a great dataset by including data from sites where detection must be severely degraded.''

We agree that this is a delicate balance. On the one hand side, we want to include as many datasets as possible to improve statistics, on the other hand, we are risking to 'contaminate' the dataset with records of too low resolution, in particular in the deeper/older part where sample resolution is decreasing for most records. Again, we have made a compromise, where we do include the lower resolution datasets all the way, but only for large eruptions, where the volcanic spike is clearly above threshold. In order to compensate for merging of several smaller spikes into a low-resolution data point, we manually check the high-resolution ECM and DEP records across the large eruptions, to see if there are several peaks involved. If there are several events occurring in one low-resolution data point we either discard that data point or we distribute the sulfate deposition into fractions based on the relative magnitude of the spikes in the high-resolution records. This procedure is now documented in the last columns of Table S3, where it is indicated if a large spike has been split and what the fractions are. We are aware that this is pushing the data a bit to the limit, but because of the large variability in sulfate deposition, we want to take advantage of all the available records in order to constrain the sulfate deposition as much as possible.

Fig S5 and line 375. Please be careful here. I believe the Rasmussen analysis makes some sense for the kind of CFA set up used by Bigler where there is a large reaction cell that is being continually mixed leading to quite a large mixing volume. In such a setup there can be a nominal data resolution of mm which is not real. The FIC setups are a bit different as they inject a slug of liquid every 2 cm (or whatever each system uses) so the resolution could never be better than that in any sense. I'm not sure your analysis really makes that distinction.

Indeed, we agree, and that is why we do the smoothing of the NGRIP CFA sulfate record to its 'effective' resolution as determined from the power spectra (as described two comments above).

Line 389. You call Fig 4 before Figs 2 and 3.

This has been changed.

Line 419. 'I found this very confusing. I am looking at Fig 2 where you show the Holocene, deglacial, stadial and interstadial. You then state that the number of eruptions is higher in cold periods when the figure shows the most in the deglacial and really no difference between the glacial and Holocene. Eventually I realised that you are only talking about stadials versus interstadials within the glacial. So firstly please make this clear throughout this section. But really is that statement true at a significant level: perhaps so in the 1year data in Fig S6, but the reader isn't looking at that, they are seeing Fig 2. You need to be much clearer that you are using Fig S6 to discuss whether the difference is an artefact and that the conclusion, as illustrated in Fig 2 is that it probably is. I realise you do say this in the end, but by pointing at Fig 2 initially you leave the reader who doesn't take the trouble to look at the supplement lost.'

Yes indeed, this point was not clear from the context. Section 4.1 is meant to be a discussion of stadial versus interstadial periods. We now state this clearly in the beginning of the section: 'This observed difference leaves us two options: either there is a true millennium-scale volcanic eruption variability related to northern hemispheric climate variability, or the dependency is an artefact caused by the very different properties of the Greenland sulfate record in stadial and interstadial periods.' and make a reference to Fig S7.

'Line 488 (2-0) Do you mean 2-0 ka? When I look at Fig 2, I don't really see a 34% increase. At least for the small (blue) category, there is no change. Perhaps describe this more precisely as to what you mean.'

Yes, we mean '2-0 ka', and yes, the increase is for the large eruptions only. This is now stated more clearly in the text as follows: 'In our 3-core Greenland sulfate deposition compilation, we also found an increased volcanic eruption frequency over the deglacial period, which is 34% higher than the late Holocene period (2-0 ka), especially for the fraction of large sulfate deposition events (Fig. 3 and Fig. 4).'.

'Line 535-544. I don't find this comparison of the ice core eruption rates against Lameve data helpful. Firstly you should also look at the analysis by Rougier et al (Rougier, J., R. S. J. Sparks, K. V. Cashman, and S. K. Brown (2018), The global magnitude–frequency relationship for large explosive volcanic eruptions, Earth planet. Sci. Lett., 482, 621-629, doi:https://doi.org/10.1016/j.epsl.2017.11.015.). It is very clear in that paper that the community is well aware that Lameve under-reports so I don't really see the value of the comparison you are making.'

Our main message is to provide an ice-core estimate of the frequency of large global volcanic eruptions during the last glacial period (Table 1). It is the first time such an estimate is possible on a long ice-core based record beyond the last 2500 yrs and we think it is quite relevant. We mention the geological record because so far that record has been the only source of information of global volcanism on those time scales. The geological record applies the VEI index that we cannot deduce from the ice core record. Instead, we compare the magnitude of the glacial eruptions to those of well-studied historical eruptions for which the VEI index is known, such as the Tambora and Salamas eruptions.

Rougier et al., 2018 analyzed volcanic frequency based on LaMEVE database and found that this volcanic record shows a deterioration of information with age, not the real variation. The GVP database (http://volcano.si.edu) has a similar problem. For the recent time, the small size volcanic eruptions are well recorded, and for the past time the large size eruptions are much easier to discover in the geological record. Furthermore, the large size volcanic eruptions show a linear pattern reflecting a stationary process, as seen for the VEI 8 index volcanoes over the past 2 Ma BP (Fig.1 in Papale et al. (2018)).

[Figure]

Caption: Cumulative plots of the number of eruptions with time for each individual VEI class. The plots represent a zoom in a (relatively) recent time region where approximately linear trends emerge. The horizontal axis refers to years Before Present (BP), conventionally set to zero at year AD 1950. Best-fit linear trends are represented by the thick red dashed lines, while the thin solid red lines bound the 95% confidence band for expected statistical variability of data23. Wider confidence limits reflect lower number of data. Each panel reports the time limit of the linear region, which generally increases with increasing VEI, and the number of eruptions in such a region, which generally decreases with increasing VEI. (Papale et al., 2018, Fig.1).

'I am unconvinced by section 4.4 Given the huge uncertainties, especially for eruptions where there is only 1 core represented in one of the ice sheets, the values are so uncertain that trying to call out individual positions in the medal table seems a bit pointless. At the least please add a column where you state how many cores are represented in each ice sheet. But I feel this section of the paper is given too much weight and should be shortened. Remember also that you certainly show no eruptions that are bipolar between 16.4 and 24.5 ka, but this is not because they were absent but only because the tie points are not good enough to know whether they are bipolar. Given this kind of issue I think the league table, while it can be included should be downplayed and put into context.'

We think there is a general interest in knowing which volcanic eruptions are the largest during the last glacial period, even if the list does not cover the complete period 0-60 ka. We now include columns in Table 2 showing in how many ice cores the individual volcanic events are identified. The majority of those large eruptions are identified in 4-6 ice cores and only a few of the lower ranking eruptions are only identified in one ice core in Greenland. Of course, the list may be improved with time as we obtain more and hopefully also more accurate estimates of the polar sulfate depositions, and of course the list will become more complete whenever we are able to include the entire Holocene and the LGM and possibly part of the earlier last glacial. But are those good arguments for not presenting a list of the largest eruptions over some 40 ka of the last glacial period investigated here for the first time in a global context? In our view, there is so much interest in this list that if we do not include it in the manuscript it will be produced by the readers, and we prefer to be the first ones to comment on it. Section 4.4 takes up less than 10 % of the manuscript text and we see it as one of the major outcomes of our work. The discussion does put our results into context in that it discusses the potential origins of several of the major eruptions that can be investigated in future studies concerned with tephra, modelling and other paleo-archives.

References:

Marshall, L., Johnson, J. S., Mann, G. W., Lee, L., Dhomse, S. S., Regayre, L., . . . Schmidt, A. (2019). Exploring How Eruption Source Parameters Affect Volcanic Radiative Forcing Using Statistical Emulation. *Journal of Geophysical Research-Atmospheres, 124*(2), 964-985. doi:10.1029/2018jd028675

Yan, J. P., Zhang, M. M., Jung, J. Y., Lin, Q., Zhao, S. H., Xu, S. Q., & Chen, L. Q. (2020). Influence on the conversion of DMS to MSA and SO42- in the Southern Ocean, Antarctica. *Atmospheric Environment, 233*. doi:10.1016/j.atmosenv.2020.117611

Zeitz, M., Levermann, A., & Winkelmann, R. (2020). Sensitivity of ice loss to uncertainty in flow law parameters in an idealized one-dimensional geometry. *Cryosphere, 14*(10), 3537-3550. doi:10.5194/tc-14-3537-2020

---

## Author Comment (AC4)

Response to the comment from Jihong Cole-Dai.

This paper by Lin and colleagues extends the records of explosive volcanism constructed systematically from polar ice cores from the Holocene into the last glacial period. This is much valuable and needed research on volcanic records and the volcanic impact on climate.

We thank you for the constructive comments.

To reconstruct volcanic records from ice cores for the glacial periods, the authors faced several daunting challenges not encountered when compiling such records for the Holocene or shorter periods. First, detection of volcanic signals in ice cores depends critically on quantifying the non-volcanic background, and the background is much more variable during the last glacial period than during the Holocene; the larger variability is the result of both climatic variations (stadials vs. interstadials) and the presence of significant non-marine biogenic sulfate. Second, the Greenland and Antarctica cores in this study were analyzed with various sampling and analytical methods, resulting in datasets of various quality. Third, due to significant layer thinning with very old ice, temporal resolution of chemical analysis is reduced in glacial ice and, as a result, further complicates the detection and quantification of volcanic signals. Fourth, significant layer thinning at much older ages makes quantitative estimation of volcanic deposition difficult. In my opinion, the authors succeeded quite well in tackling these thorny issues and came up with a remarkable record of large volcanic eruptions for the last 60,000 years. It is worth noting that, due to the highly variable background sulfate levels during the glacial part of the period covered in this study, only extremely large eruptions are detected and quantified. Nonetheless, a record of extremely large eruptions is very valuable when it comes to assessing the climate impact of explosive volcanism, for we know from other studies that very large eruptions exert the most significant impact on climate.

The conclusions of this study are significant, not only because it is the first time that a systematic study of volcanism during the last 60 ka yields a robust record, but also it demonstrates that ice cores are capable of providing valuable information regarding volcanism and its climatic impact on time scales of millennia and longer, supplementing and/or enhancing knowledge from geological records.

I would like to offer a comment on a technical aspect. I find that in several places in this paper, the authors use terms such as "measurement techniques" and "measurement methods" for how ice core analysis yields sulfate data and how different "techniques" or "methods" yield data of various quality. Regarding the chemical measurement of sulfate, there are only two techniques: ion chromatography (IC) and mass spectrometry (ICP-MS for sulfur). The quality of data from these techniques is the same or similar, for any ice core samples. Where data quality may vary is when different sampling methods are used: discrete, continuous melting with online IC or ICP-MS measurement, or continuous melting followed by off-line IC measurement. Often, these sampling methods determine measurement resolution or temporal resolution. For example, when sulfate was measured using IC on discrete samples, the resolution is lower than that when the measurement was using FIC on samples from a melter, due to the higher sampling resolution of the melter. The quality of sulfate data is the same, as sulfate in both cases were measured with ion chromatography.

We thank you for those insights. However, in this study we apply the NGRIP sulfate record that is obtained using a continuous absorption method that is different from the IC and ICP-MS techniques. The method is introduced in Röthlisberger et al. 2000 and applied by Bigler et al. (2002) to measure sulfate and detect volcanic signals. We now include the references in the manuscript.

In addition, I ask the authors to consider the following comments on specific passages in the paper.

Line 68. Small deposition at low accumulation sites could be also, or even mostly, due to reduced wet deposition.

This has been changed. In the manuscript it now says: Another reason for the lateral sulfate deposition variability is the amount and patchiness of snowfall, which may locally enhance the sulfate deposition for high snowfall areas compared to low snowfall areas for a volcanic event. Moreover, there may be more absent sulfate deposition events caused by post depositional processes on the snow surface, such as wind erosion (Gautier et al., 2016).'.

Line 88-91. Are you saying that uncertainty in thinning-rate estimates also contributes to uncertainty/variability of volcanic deposition?

Yes, there are uncertainties on the applied, modelled thinning functions that are not well quantified, and in the deeper ice the thinning is very significant potentially leading to a large error of the estimated sulfate deposition.

Line 113. In my opinion, the main problem with tephra ID is not that tephra does not deposit with sulfate simultaneously. The difference in timing of tephra and sulfate deposition is usually small, less than one year. The main problem is that it is impossible to perform continuous search and analysis of tephra in any deep/long ice core with the current analytical (or technological) tools. Additionally, there are no objective standards on matching the tephra in ice core to ash from a particular eruption.

We have changed the text as follows: 'However, tephra layers are not always associated with sulfate peaks and most volcanic sulfate signals have no tephra associated with them.'.

Line 150. The word "analytical" should be inserted in "different methods".

This has been changed.

Line 166-169. MSA correction is needed only when sulfur, not sulfate, was measured. In fact, the correction should not or cannot be applied to sulfate data.

We agree. This was a mistake and the MSA correction is only mentioned for the WDC record.

Line 220. Is there such an assumption?

No, we just want to mention that the meterological conditions are different between the last glacial period and the Holocene period, so that some of the knowledge we have from the Holocene period may not we valid in the last glacial.

Line 294. "sulfate deposition strongly influenced by sulfate record type (IC, FIC, CFA)"? See my comment on a technical aspect.

See our earlier reply.

Lines 301-304. Deposition variation is due to different sampling methods, not measurement methods or techniques. See my comment on a technical aspect.

Again, in this case the measurement methods are different. Fig.S3 shows the volcanic sulfate deposition as obtained by different analytical techniques. More explanations are provided above.

Line 350. "Two bipolar volcanic eruptions are known from tephra to be Icelandic origin: Could signals in Antarctica ice cores be from contemporaneous eruptions in SH? I notice that the authors use the term "bipolar volcanic eruptions" for contemporaneous signals in both Greenland and Antarctica ice cores. I think it is acceptable to use this term, with the caveat that such contemporaneous signals may be left by simultaneous eruptions in both hemispheres, rather than a single eruption.

Yes, there is of course a risk that some of the bipolar synchronization events are not bipolar, but rather 'false' coincidences of eruptions occurring almost simultaneous in the two Hemispheres. We make the assumption for the bipolar volcanic eruptions that they are one and the same event based on our well synchronized volcanic sulfate signals from Greenland and Antarctic ice cores. However, based on the sulfate concentration alone we cannot exclude contemporaneous SH volcanoes, and in view of the rather low volcanic frequency the likelihood of co-occurring NH and SH volcanoes is very small. We are addressing this issue in Lines 131-134: ' It has recently become possible to test if sulfate has indeed reached the stratosphere, which is a prerequisite for being globally distributed, as the sulfate undergoes characteristic isotope fractionation in the stratosphere (Burke et al., 2019; Gautier et al., 2018; Crick et al., 2021; Baroni et al., 2008), but these analyses are still scarce for the Glacial.'

Reference

Röthlisberger, R., Bigler, M., Hutterli, M., Sommer, S., Stauffer, B., Junghans, H. G., and Wagenbach, D.: Technique for continuous high-resolution analysis of trace substances in firn and ice cores, Environmental Science & Technology, 34, 338-342, 2000.

Bigler, M., D. Wagenbach, H. Fischer, J. Kipfstuhl, H. Millar, S. Sommer & B. Stauffer. 2002. Sulphate record from a northeast Greenland ice core over the last 1200 years based on continuous flow analysis. In Annals of Glaciology, Vol 35, ed. E. W. Wolff, 250-256.

---

## Author Comment (AC5)

Response to the comments from Thomas Aubry, Lauren Marshall and Anja Schmidt:

'This manuscript by Jiamei Lin and co-authors represents the first effort to constrain stratospheric volcanic SO2 emissions for the 60-9 ka period using a bipolar array of ice cores, and these emissions are then used to estimate the corresponding volcanic forcing. This will without doubt be a very useful contribution for the community working on volcano-climate interactions.'

We are grateful for the positive comments and suggestions from Thomas Aubry, Lauren Marshall and Anja Schmidt. Below, we provide our responses in blue color.

'We would like to draw the attention of the authors to potential improvements for estimating volcanic forcing from emissions.

First, to estimate a global-mean Stratospheric Aerosol Optical Depth (SAOD), the authors use a linear scaling between SAOD and the aerosol loading. However, it is well known that for large eruptions this relationship is not linear (e.g. Crowley and Unterman, 2013). As highlighted by the authors, the scaling used in their work is calibrated against the 1991 Mt. Pinatubo eruption and the reference used does not employ the latest estimates of SO2 mass and SAOD for this eruption. For example, the post-Pinatubo peak global mean SAOD in Crowley and Unterman (2013) (ca. 0.14-0.15) is 16% larger than in the GloSSAC dataset (0.12-0.13, Kovilakam et al. 2020). We suggest that the authors consider either using the EVA model (Toohey et al., 2016) or the EVA_H model (Aubry et al., 2020) to obtain SAOD. EVA is calibrated using more up-to-date data for Pinatubo and is also a reference model for the community as it has been used to derive the volcanic forcing for CMIP6's Paleoclimate Model Intercomparison Project (PMIP4). EVA_H is an extension to EVA that was calibrated using the full 1979-2015 period with state-of-the-art observational datasets. Additionally, in EVA_H the predicted global mean SAOD depends on the eruption latitude, which is not the case in EVA.

We thank the authors for pointing out potential limitations of our method for eruptions with excessive sulfate loadings and recommend the easy volcanic aerosol (EVA) model (Toohey et al., 2016) and the EVA_H model (Aubry et al., 2019) to derive SAOD from the stratospheric sulfate aerosol loading. We now apply those scaling factors in addition to the approach of the submitted version of the manuscript (see below).

Second, to convert global-mean SAOD to global-mean radiative forcing, the authors use the scaling factor of Hansen et al. (2005). This scaling factor was constrained using climate model simulations for the 1991 Mt. Pinatubo eruption without full consideration of rapid adjustments. Several recent studies have suggested that consideration of rapid adjustments leads to a reduction in the scaling factor (e.g., Gregory et al., 2016; Larson & Portmann, 2016; Schmidt et al., 2018; Marshall et al., 2020). Revised scaling factors for a wide range of eruptions are available in Marshall et al. (2020). Collectively, these studies suggest a reduced conversion factor compared to Hansen et al. (2005) and IPCC AR5.

To obtain the global radiative forcing, we now also adopt the revised scaling factor from Marshall et al., (2020).

We acknowledge that using more recent methods will result in differences in reconstructed forcings that are likely small relative to uncertainties in ice-core derived estimates of the SO2 mass. We nonetheless think that it remains important to acknowledge and use the latest tools developed by the community to provide volcanic forcing estimates. At the minimum, the authors should discuss differences that may emerge from using different scaling factors.

Thank you for recommending the newer approaches to reconstruct the volcanic forcing. We have added to following text to section 4.3: 'To estimate the volcanic radiative forcing from eruptions occurring in the last glacial and early Holocene we need to constrain the sulfate stratospheric aerosol loading (we applied the method of Gao et al. (2007)), to convert the stratospheric aerosol loading into the global mean stratospheric aerosol optical depth (SAOD) (we applied the methods of Crowley and Unterman. (2013) and that of Aubry et al. (2020)), and to convert global mean SAOD to the global mean radiative forcing (we applied the methods of Hansen et al. (2005) and that of Marshall et al. (2020)). The global stratospheric sulfate aerosol loading requires a separation of NH high-latitude eruptions from other eruptions, as the two eruption groups are scaled differently (Gao et al., 2007). We defined NH high latitude eruption as eruptions that occurred at a latitude above 40° N. To identify the NH high latitude eruptions, we applied a Support Vector Machine learning classifier model (SVM – see methods section), that is trained by the bipolar sulfate deposition of volcanic

eruptions for which the eruption site is known. We applied 17 Holocene and 4 glacial volcanic eruptions of known origin (Table S6) to predict that 50 out of 85 bipolar eruptions of unknown origin are likely to have occurred in the NH high latitudes (Fig. 1 and Fig. 6). We then reconstruct the volcanic radiative forcing using three different approaches:

1) The global mean SAOD is obtained using the method of Crowley and Unterman. (2013) and the radiative forcing calculation applies the scaling factor of Hansen et al. (2005). Here the volcanic radiative forcing is calibrated against Pinatubo 1991 AD (at 15°N), and the approach is similar as Sigl et al. (2015).

2) The global mean SAOD is obtained using the scaling factor of Aubry et al. (2020) and the radiative forcing calculation applies the scaling factor of Hansen et al. (2005). This approach is similar as the one used in the IPCC AR5.

3) The global mean SAOD is obtained using the scaling factor of Aubry et al. (2020) and the radiative forcing calculation applies the scaling factor of Marshall et al. (2020), which considers rapid aerosol adjustment for large volcanic eruptions.

All of the reconstructed volcanic radiative forcings are calibrated and evaluated based on modern volcanic eruptions, and they are therefore potentially biased when applied to the eruptions occurring in the very different glacial climate. Table S5 and Fig. S11 present the reconstructed volcanic radiative forcing of individual volcanic events using three approaches. The reconstructed volcanic forcing obtained by method 2) and 3) is significantly weaker than that obtained by method 1) by a factor of 1.3 and 2.8, respectively, when integrated over all events. In the following, we adopt method 3) to present the reconstructed volcanic forcing values.'.

The figure below shows the comparison of the above three volcanic forcing reconstructions. The reconstructed volcanic forcing by the recent methods (Y axis values in red and blue dots) is relatively weaker than that reconstructed by the same method as Sigl et al. (2015) (X axis values in red and blue dots). The total volcanic forcing value for all the bipolar volcanoes reconstructed by the second and the third methods are respectively 1.3 times and 2.8 times weaker than that using the same method of Sigl et al. (2015).

[Figure]

Thanks again for a very interesting manuscript.

Thank you very much for your comments that have led to an improved manuscript.

---

## Author Response (AR1)

Dear editor and editorial support team

We have uploaded the response to the reviews and tracked the changes with mark-up (insertions in underline and deletions in strikethrough) in author's track change file.

If you find any questions, please no hesitate to contact us.

Best regards, all the authors

---

## Author Response (AR2)

Dear editor and reviewers,

Thank you for your fast review in the second round and the comments to revise the manuscript. We corrected the manuscript according to the specific comments from the reviewer reports in the revised version that is marked up and uploaded.

Concerning the SVM method to estimate the eruption latitudinal band, we rewrote the section as shown below. Please let us know if the text has improved.

Best regards, the authors

Point-to-point replies:

*All three reviewers are generally happy with the revised document, I thank you for your work in addressing the the comments of the first round. All three reviewers did however point out a few technical/minor corrections, which I ask you to correct or address before final publication. Please see the reviewer reports for the specific comments.*
*The most significant comment is from reviewer 1 and is related to the use of SVM to estimate the eruption latitude range. I do agree that it would be useful to include a little more detail on the method and result. For example, how does the resulting distribution (NH/Tropics/SH) compare to other reconstructions (e.g., for the past 2500 years)? A sentence or two on the strengths and weaknesses of your new method would also be welcome.*

The text now looks as follows: 'The volcanic sulfate deposition in Greenland and Antarctica shows a distribution pattern related to the latitudinal band of the eruption site (Fig. 1) (Marshall et al., 2019). To estimate the latitudinal band of bipolar volcanic eruptions of unknown origin, we applied the Support Vector Machine (SVM) classification model of Hastie et al. (2009) and Vapnik (1998) that is based on a kernel function generation and logistic regression. The model was trained using 4 eruptions from the last glacial period and 17 eruptions from the last two millennia for which the eruption site is known from the deposit tephra in the ice (Fig. 1(b) and Table S6). The input values of each eruption for the model on the training set are the average Greenland sulfate deposition, the average Antarctic sulfate deposition and the latitudinal band of the eruption site (above 40°N, 40°N-40°S, or below 40°S). The cross-validation is performed on the training set consisting of 10% of the total training eruptions selected at random. Then, this trained model is applied to give a best estimate of the latitudinal band of the bipolar eruptions for which the eruption site is unknown. The model output parameters - kernel scale, box constraints and Bayesian optimization - show that the model has good performance (Fig. S4). Due to the low number of known volcanoes erupted in the high latitudes of the Southern Hemisphere, the method does not allow unambiguous identification of eruptions potentially located in this region. The bipolar eruptions of unknown origin are thus predicted into two latitudinal bands – above 40°N (NHHL) and below 40°N (LL or SH) (Table S5). The latitudinal band assignment for the four bipolar eruptions at the onset of the Younger Drays period (Table S5) is similar to that expected from comparing the relative sulfate deposition in Greenland and Antarctica (Abbott et al., 2021). A weakness of the method is that the training set mostly consists of volcanic eruptions for which the sulfate deposition is much smaller than that of the large eruptions occurring during the last glacial period. Details of the SVM method are provided in Hastie et al. (2009), page 17.'.

*Some other technical comments specifically related to this aspect of the work:*
*1. At line 310, you mention the 21 eruptions used to train the SVM. You reference table S6, but these are also shown in Fig 1b, correct? If so, it would be useful to refer to the figure also.*

Indeed, done.

*2. At line 312 of the tracked changes document, "The cross-validation used for tuning the algorithm is 10-fold partition for each evolution." does not make any sense to me. Can this be made clearer?*

Removed.

*3. I was surprised to see the Unidentified large event at 38.1 kaBP on Figure 6 plotted as a SH extratropical event, since the description of the SVM method said it can only assign either north or south of 40N. At around line 645, it appears this event is suggested to likely be a low latitude event, while in the caption for figure 6 it is said to be "very likely from Southern Hemisphere". Perhaps I have missed some discussion elsewhere but this seems to be a contradiction.*

Thank you for picking this error up. Indeed, the 38.1 ka eruption is very likely to be tropical. It is the 46.68 ka event that has two times higher sulfate deposition in Antarctica than in Greenland that we wanted to mention as likely originating in the SH. The figure is now corrected.

*4. The caption for Fig 6 says: "Volcanoes that erupted sites are predict above 40°N are marked at 40°N", however, on the plot, those events seem to be plotted at around 50°N.*

Corrected.